# Taxonomic and environmental distribution of bacterial amino acid auxotrophies

Josep Ramoneda [1] ✉, Thomas B. N. Jensen [1,2], Morgan N. Price [3], Emilio O. Casamayor [4] & Noah Fierer [1,5] ✉

Many microorganisms are auxotrophic—unable to synthesize the compounds they require for growth. With this work, we quantify the prevalence of amino acid auxotrophies across a broad diversity of bacteria and habitats. We predicted the amino acid biosynthetic capabilities of 26,277 unique bacterial genomes spanning 12 phyla using a metabolic pathway model validated with empirical data. Amino acid auxotrophy is widespread across bacterial phyla, but we conservatively estimate that the majority of taxa (78.4%) are able to synthesize all amino acids. Our estimates indicate that amino acid auxotrophies are more prevalent among obligate intracellular parasites and in free-living taxa with genomic attributes characteristic of 'streamlined' life history strategies. We predicted the amino acid biosynthetic capabilities of bacterial communities found in 12 unique habitats to investigate environmental associations with auxotrophy, using data compiled from 3813 samples spanning major aquatic, terrestrial, and engineered environments. Auxotrophic taxa were more abundant in host-associated environments (including the human oral cavity and gut) and in fermented food products, with auxotrophic taxa being relatively rare in soil and aquatic systems. Overall, this work contributes to a more complete understanding of amino acid auxotrophy across the bacterial tree of life and the ecological contexts in which auxotrophy can be a successful strategy.

Microbial auxotrophy (i.e. the inability of microorganisms to synthesize the compounds they require for growth) has been identified in taxa isolated from many environments[1–11]. The loss of biosynthetic genes can, under certain conditions, confer a selective advantage due to the corresponding reduction in metabolic and energetic costs[12–14]. Auxotrophy can be particularly advantageous when the essential metabolites can be readily obtained from the surrounding environment, or from nearby cells, leading to the expectation that in environments with abundant nutrients or close-range interactions, auxotrophy will be an adaptive trait[15]. For example, under laboratory conditions, *E. coli* supplied with amino acids can evolve amino acid auxotrophies in under 2000 generations and outcompete its ancestral prototrophic relatives (i.e. taxa that have the ability to synthesize all amino acids[16]). Another example are obligate intracellular parasites, which have among the smallest genomes of all bacteria and are commonly auxotrophic for vitamins and certain amino acids available from their host[17].

Microorganisms can be auxotrophic for multiple types of metabolites. The most frequent auxotrophies are those for vitamins[3,6,18–21], amino acids[22–27], and diverse cofactors (e.g. heme groups[28]). Here, we focus on amino acid auxotrophies because amino acids can be important both as energy sources and as building blocks of the

[1]Cooperative Institute for Research in Environmental Sciences (CIRES), University of Colorado, Boulder, CO, USA. [2]Center for Microbial Communities, Department of Chemistry and Bioscience, Aalborg University, Aalborg, Denmark. [3]Environmental Genomics and Systems Biology, Lawrence Berkeley National Laboratory, Berkeley, CA, USA. [4]Spanish Research Council (CSIC), Center for Advanced Studies of Blanes (CEAB), Blanes, Spain. [5]Department of Ecology and Evolutionary Biology, University of Colorado, Boulder, CO, USA. ✉e-mail: ramoneda.massague@gmail.com; noah.fierer@colorado.edu

proteome, the costs associated with synthesizing amino acids are reasonably well-constrained[12], and because the amino acid biosynthetic capabilities of many bacteria can be inferred with recent improvements in our understanding of biosynthetic pathways and the bioinformatic tools to infer amino acid auxotrophies[29–32]. In synthetic assemblages, amino acid cross-feeding can be an ecologically stable strategy when interacting partners complement each other in their metabolic capabilities[33]. Thus, it is often assumed that auxotrophic interactions and the cross-feeding of amino acids are a key factor structuring microbial communities[15]. While there is limited evidence for auxotrophy-mediated amino acid exchange in microbial communities found in natural systems, previous work has suggested that this phenomenon likely occurs in microbial consortia responsible for hydrocarbon degradation[8], methanogenesis[34], and anammox[35].

Auxotrophy is expected to be more common in habitats where the essential metabolites are more readily available and diffusible. For example, protein-rich environments such as dairy products contain a high availability of amino acids[36], and are dominated by well-known amino acid auxotrophs such as bacteria from the genus *Lactobacillus*[24]. The physical structure of microbial habitats can also influence the availability of essential metabolites. Auxotrophies may be particularly prevalent among bacteria living in biofilms or in well-mixed systems, where metabolites can more readily be exchanged between taxa primarily due to their spatial proximity[37,38]. Generally, we would expect that communities from different environments should vary with respect to the prevalence of auxotrophies due to differences in the amounts and types of metabolites available. For example, we would expect bacterial amino acid auxotrophs to be more common in host-associated systems where amino acid availability is reasonably high, such as the human gut[22,26,27]. However, the broader prevalence of auxotrophic bacteria in other types of microbial systems (including soil and aquatic systems) remains largely undetermined.

Using genomic information alone, it is possible to predict the metabolic capabilities of many bacterial taxa[31,32,39–41]. These metabolic pathway models rely on a priori knowledge of the genes involved in the metabolic pathways of interest and allow for the prediction of auxotrophy in any taxon for which high quality genomic information is available. For example, D'Souza et al.[13] used genomic information from 949 full genomes to estimate that 76% of bacterial taxa were auxotrophic for at least one essential metabolite. The frequent application of metabolic pathway models contrasts with the paucity of experiments that empirically validate the predictions of these models. The experimental validation of auxotrophy typically requires challenging and time consuming in vitro assays that are, by definition, difficult to conduct on the large fraction of bacterial taxa that remain uncultured[42]. Those studies that have attempted to empirically validate predictions of auxotrophy show that genome-based models largely underestimate the metabolic capabilities of bacterial taxa[29–31,43]. For example, Price et al.[29] studied 10 bacterial genera that were predicted to be auxotrophic for several amino acids, but found that these taxa could grow on minimal media in the absence of externally supplied amino acids. Using genome-wide mutant fitness data, the authors identified genes for 9 of the 11 missing steps in amino acid biosynthesis. While many biosynthetic pathways remain poorly understood[44], new empirical findings and conservative bioinformatic approaches make it possible to infer bacterial auxotrophies[31,32,43].

Here, we predicted the prevalence of amino acid auxotrophies across a broad diversity of bacteria by analyzing 26,277 genomes representing 12 different bacterial phyla. We also compared the predicted prevalence of amino acid auxotrophies from 13,523 representative taxa found in 12 different habitats, ranging from soils, freshwater, and marine waters, to engineered systems such as activated sludge and food products, and to host-associated systems including the human gut, skin, and plant leaf surfaces. We validated the predictions of a metabolic pathway model of bacterial auxotrophy[31] by

compiling empirical information on the metabolic capabilities of diverse bacterial taxa to minimize the overestimation of auxotrophy. Finally, we evaluated which genomic features are more frequently associated with bacterial amino acid auxotrophy to characterize the broader life history strategies that differentiate amino acid auxotrophs from prototrophs. By covering a broad range of taxa and habitats we provide a comprehensive view on the taxonomic and environmental signatures of amino acid auxotrophies in bacteria.

## Results and discussion
### Model validation
To test our ability to infer amino acid auxotrophy from genomic analyses, we first validated our model after predicting the amino acid biosynthesis capabilities of 171 taxa that can make all amino acids (prototrophs). Doing so allowed us to quantify how many genes need to be missing from an amino acid biosynthesis pathway in a certain organism to be considered auxotrophic for that amino acid. To minimize the overestimation of auxotrophy, we found that at least 40% of the genes needed to be missing in a given amino acid biosynthesis pathway to obtain a very low 0.4% rate of false positives (i.e. erroneously predicted auxotrophies). This means that our model predictions were correct in ~99% of the cases in which an organism was able to synthesize a given amino acid. Only for serine and cysteine (4% error) did our model incorrectly predict amino acid auxotrophies (i.e. inferring auxotrophies when the taxa were actually capable of synthesizing those amino acids, Supplementary Fig. 1). In the case of serine, 6 of the 7 genomes that were misclassified as auxotrophic belonged to taxa from the phylum Desulfobacteria, which are typically sulfate-reducers (the remaining genome belonged to a green sulfur bacterium from the Chlorobiaceae, Bacteroidetes; Supplementary Data 1). A group of sulfate-reducing bacteria, including Desulfovibrio and related genera, appear to produce serine from pyruvate or related compounds as in the standard pathway[45], but the genes involved are not known. The phylum Desulfobacteria was not included in the analyses presented below. Similarly, all the genomes that were misclassified as cysteine auxotrophs belonged to phyla not included in this study such as the Desulfobacteria and the Aquificae, also characterized by having sulfur-related metabolisms (Supplementary Data 1). We found that these genomes contained the cysteine synthase gene (*cysK*), which makes it unlikely that these taxa synthesize cysteine via alternative pathways. Together, these results suggest that our decision to require at least 40% of the genes to be missing to infer auxotrophies for cysteine and serine auxotrophy primarily affected less abundant phyla not included in the study.

We then quantified the rate of false negatives (i.e. inferring prototrophy for amino acids that taxa cannot synthesize) using genomes from taxa with experimentally determined auxotrophies compiled from the literature (Supplementary Table 1). Applying our threshold that a minimum of 40% of genes from a pathway had to be missing to consider a genome auxotrophic for a given amino acid led to false negative rate of 20% (i.e. the proportion of amino acids in each genome for which our model predicted taxa to be prototrophic when they were auxotrophic, Supplementary Fig. 2). On a per genome basis (i.e. predicting whether a given genome is auxotrophic for 1 or more amino acids versus prototrophic), our model correctly infers prototrophy in 93% of the cases, and infers that a taxon is auxotrophic for at least 1 amino acid correctly in 95% of the cases. This means that, although the model tends to underestimate the number of amino acids that a given taxon is unable to synthesize, we can accurately identify when a taxon is generally auxotrophic or prototrophic. We recognize that our current understanding of amino acid biosynthesis pathway derives from taxa that have been cultured, and that improved knowledge beyond those taxa is required to improve our inferences of auxotrophies in particular groups.

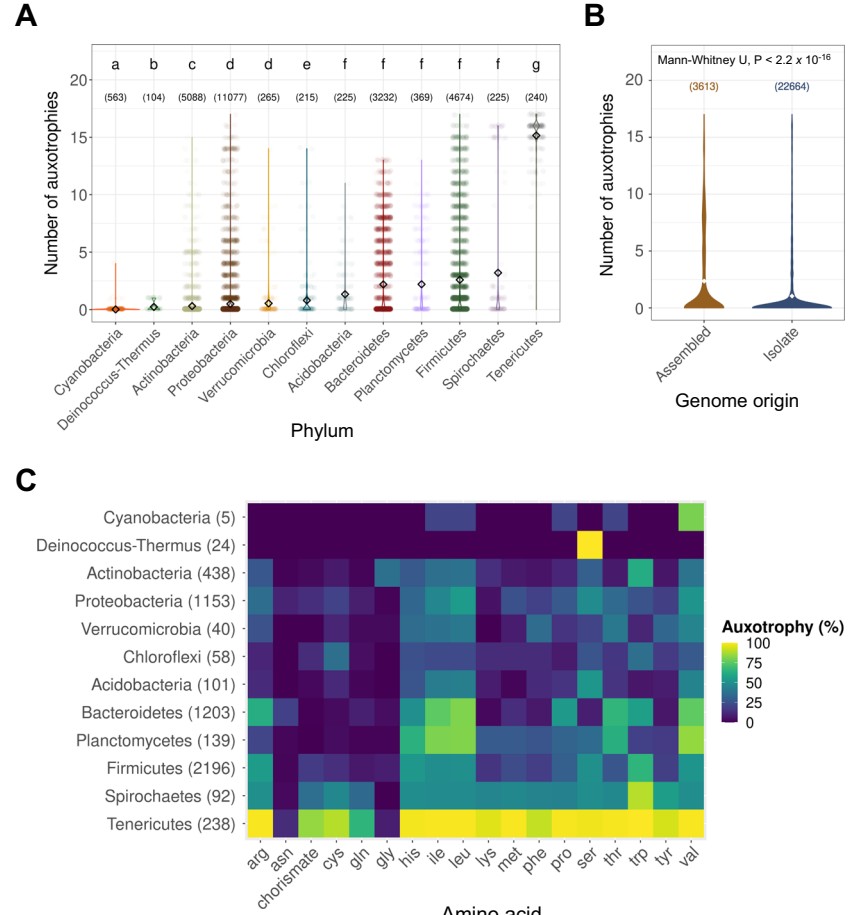

**Fig. 1 | Amino acid auxotrophy across the predominant bacterial phyla.**
**A** Prevalence of amino acid auxotrophy in bacterial taxa from the most common phyla (*N* = 26,277 genomes). **B** Prevalence of amino acid auxotrophy in genomes derived from environmental metagenomes (MAGs) or single cells (SAGs) (Assembled), and in genomes obtained from bacterial isolates (Isolate). The mean number of amino acid auxotrophies in **A**, **B** is indicated with white diamonds. **C** Proportion of taxa that are auxotrophic for each of the 17 amino acids and chorismate out of the total number of auxotrophic taxa (*N* = 3613 genomes). Numbers in brackets in panel A indicate the number of genomes for which we predicted amino acid auxotrophy, numbers in brackets in panel B indicate the number of assembled and isolate genomes included, and numbers in brackets in **C** indicate the subset of taxa within each phylum that were predicted to be auxotrophs for at least one amino acid. Letters in panel A indicate statistical differences (*P* < 0.05) between phyla based on Mann–Whitney U tests with *P*-values Bonferroni-corrected for multiple comparisons.

Previous genome-based studies have largely overestimated amino acid auxotrophy, despite mounting evidence that most of these inaccurate predictions come from knowledge gaps or from lack of awareness of alternative biosynthetic pathways[29,30]. A number of studies have used focused culturing efforts to identify auxotrophies in experimental isolates[24,25,29,32,46] and high-throughput culturing techniques make it possible to screen for bacterial growth across a wide range of media types[47,48]. We recognize that our approach likely misses a number of auxotrophies, but it does provide a more conservative perspective on the actual amino acid biosynthesis capabilities of most bacterial taxa. The fact that we only found 19 taxa with genomic data available and known amino acid auxotrophy profiles highlights the difficulties of conducting in vitro experiments to confirm amino acid auxotrophies[32]. Future work could benefit from advances in high-throughput cultivation-based approaches to experimentally identify auxotrophies[49] and expand the datasets needed for validation of genome-based models[50]. Dedicated efforts combining extensive media testing, whole genome sequencing, and comparative genomics will further reduce uncertainty around amino acid biosynthesis in bacteria. Until then, we are confident that our approach is conservative, recognizing that we are likely underestimating the occurrence of some amino acid auxotrophies.

## Prevalence of amino acid auxotrophies in bacteria

We used our genome-based approach to predict amino acid auxotrophies in 26,277 bacterial taxa from the 12 phyla with >100 non-chimeric representative genomes estimated to be >95% complete in the Genome Taxonomy Database (GTDB, release 207)[51]. A large majority of taxa (78.4%), each represented by a single genome, were inferred to be able to synthesize all amino acids (i.e. were completely prototrophic; Fig. 1A). This prediction contrasts with the previous comprehensive study of amino acid auxotrophy in bacteria, which was based on 949 sequenced genomes with the authors reporting that only 24% of bacterial taxa were able to synthesize all amino acids[13]. There are many reasons this discrepancy may exist, but it does suggest that the GapMind predictive framework applied here yields a more conservative estimate of amino acid auxotrophies (as explained above) and is less likely to incorrectly infer auxotrophies when specific biosynthetic genes are not detected in genomes.

Even though our model estimated that 78.4% of the 26,277 bacterial taxa were deemed to be completely prototrophic, there was a high degree of variation in the distribution of amino acid auxotrophies across bacterial taxa. We observed the lowest proportion of auxotrophs in the Cyanobacteria (0.9%) and the highest proportion in the Tenericutes (99.2%). The phyla with the largest numbers of representative genomes all contained large numbers of both auxotrophs

and prototrophs, with members of the Actinobacteria (8.6%) and Proteobacteria (10.4%) having significantly lower proportions of auxotrophs than Bacteroidetes (37.2%) and Firmicutes (37.0%) (Fig. 1A; Supplementary Table 2). Our finding that the Bacteroidetes and Firmicutes phyla contain higher proportions of auxotrophs than most other phyla is in agreement with previous work[13,15,30,52]. Similarly, our finding that most Cyanobacteria are prototrophic for all amino acids is in line with previous work suggesting that Cyanobacteria are able to synthesize all amino acids[53] and our observation that only 0.8% of the Tenericutes are prototrophic is to be expected given that auxotrophies are widely observed in this group, which is mostly represented by obligate, intracellular parasites[54,55].

Our analysis of the prevalence of auxotrophies at the family level emphasizes the broad taxonomic distribution of auxotrophs. We predicted the prevalence and identity of amino acid auxotrophies across the predominant bacterial families (51 families from the 12 phyla with at least 100 available genomes; Supplementary Fig. 3). Less than a quarter (21.6%) of the families contained more auxotrophic than prototrophic taxa. The Mycoplasmataceae was the only family where all bacterial members were predicted to be auxotrophs, as expected for this group of intracellular parasites that obtain required nutrients from their host[56]. All families where over 80% of their members were predicted to be auxotrophs contained predominantly host-associated taxa, including Coriobacteriaceae[57], Lactobacillaceae[58], and Streptococcaceae[59] (Supplementary Fig. 3). On the opposite end of the spectrum, 54.9% of the 51 families had less than 10% auxotrophic taxa (Supplementary Fig. 3). The least auxotrophic families were the Streptomycetaceae (0.1%), Paenibacillaceae (0.2%), and the Pseudomonadaceae (0.3%).

## Associations between amino acid auxotrophy, genome size, and genome origin

We found that the prevalence of auxotrophs was significantly lower for genomes derived from bacterial isolates compared to those genomes assembled from environmental metagenomes (MAGs) and single cells (SAGs) (Mann-Whitney U, $p < 0.001$; Fig. 1B). Note that all MAG/SAG genomes included in the study were thoroughly filtered for completeness (>95% complete), absence of chimerism, and were required to contain an assembled 16 S rRNA gene. We also found that MAGs/SAGs, >95% of which represent uncultivated taxa, had generally smaller genomes and higher predicted minimal doubling times than genomes derived from cultured isolates (Welch two-sample t-test, $p < 0.001$; Supplementary Fig. 4A, B), in agreement with previous findings[60]. Crucially, the number of amino acids that taxa were unable to synthesize was inversely proportional to their genome size ($r = -0.40$, $p < 0.001$; Supplementary Fig. 4C). This general negative association between genome size and auxotrophy across phyla suggests that the higher number of auxotrophies observed in MAGs/SAGs is likely due to evolutionary processes associated with genome size reduction, and not potential annotation or completeness biases. Isolate-derived genomes had higher completeness (99.2% average completeness) than those from MAGs/SAGs (97.6%), but this difference alone is likely insufficient to result in a sizeable difference in the number of estimated auxotrophies. We also verified that the potential impact of genome completeness on predicted amino acid auxotrophy was minor based on the weak correlation between genome completeness and the number of auxotrophies per genome (within MAGs/SAGs $r = -0.07$; within isolates $r = -0.14$). We also verified that the phyla with the highest proportions of auxotrophic taxa did not typically contain a larger proportion of MAG/SAG genomes (Supplementary Fig. 5). These results suggest that many bacterial taxa are not readily cultivated because they have life history strategies characterized by slow growth and complex external nutrient requirements that impair growth under laboratory conditions[42]. This seems unsurprising as phyla with low proportions of auxotrophs (e.g. Cyanobacteria or Actinobacteria) tend

to have larger genomes compared to phyla with higher proportions of auxotrophs[61], and genome reduction by loss of biosynthetic genes has previously been associated with auxotrophy across bacterial groups[62] (see below for further discussion of this point).

## Amino acid auxotrophies associated with specific bacterial phyla

We next investigated which specific amino acid auxotrophies were most common across bacteria. Auxotrophic bacteria were most frequently auxotrophic for leucine (58.5%), valine (57.8%), and isoleucine (54.9%) (branched-chain amino acids), and were the least likely to be auxotrophic for asparagine (7.0%), glycine (7.2%), and glutamine (9.3%) (Fig. 1C). The availability of branched-chain amino acids controls the virulence gene expression in diverse host-associated bacteria, and auxotrophy for these amino acids has been suggested to be an adaptation to regulate bacterial metabolic activity with changes in external nutrient levels[63]. Generally, the amino acid auxotrophic profiles were primarily dictated by the identity of the amino acids rather than the taxonomic affiliation of the genomes in question, meaning that most phyla were more auxotrophic for the same amino acids (Fig. 1C). There were some exceptions to this pattern. For example, in the Actinobacteria (91.4% prototrophs) 61.6% of the auxotrophic taxa could not synthesize tryptophan (Fig. 1C). Notably, 41.6% of those actinobacterial tryptophan auxotrophs belonged to the gut-associated genera Collinsella and Olsenella[64]. We verified that the number of genes in a given amino acid biosynthesis pathway was not strongly correlated with the proportion of auxotrophic taxa for that amino acid ($r = -0,43$, $p = 0.100$). Note that the predicted auxotrophy for serine in the Deinococcus-Thermus phylum is likely due to a novel phosphoserine phosphatase in Thermus thermophilus, which has not been incorporated into GapMind[65].

In contrast to previous studies, we did not find a significant correlation between the proportion of auxotrophic taxa for each amino acid and the metabolic cost of each amino acid calculated from the number of P-bonds required to synthesize a given amino acid ($r = -0.24$, $p = 0.4$; Supplementary Fig. 6A)[12]. When we explored this relationship within each of the predominant phyla, we only found a significant correlation in the phylum Spirochaetes ($r = 0.71$, $p = 0.001$; Supplementary Fig. 6B).

## Prevalence of amino acid auxotrophy across habitats

We analyzed representative genomes from bacterial taxa found across 12 different habitats to assess general patterns in amino acid auxotrophies (Table 1). The habitats included in our analyses covered a broad range of habitat types, including terrestrial (bulk soil, rhizosphere soil), aquatic (freshwater lakes, marine surface waters), engineered (activated sludge and residential plumbing), host-associated habitats (phyllosphere, human gut, human skin, and human oral cavity), and fermented foods (cheese and sourdough). We identified between 148 (cheese) and 2949 (phyllosphere) representative genomes per habitat (13,523 genomes in total) (Table 1, see Methods). The proportion of taxa that were capable of synthesizing all amino acids was highly variable across habitats. More than 95% of bacteria found in rhizosphere soils, residential plumbing, and bulk soils were capable of synthesizing all amino acids (Fig. 2A; Table 1). In contrast, less than half of the bacteria in the human gut (41.6%) and oral cavity (24.7%) were prototrophic for all amino acids (Fig. 2A). The habitat-specific patterns in auxotrophy prevalence were still evident even when we restricted our analyses to the phylum Proteobacteria, the most ubiquitous phylum across habitats and a phylum with biosynthetic pathways that have been relatively well-studied[31]. These proteobacterial-specific analyses also show that the human gut and oral cavity were inferred to have the highest proportions of auxotrophic taxa (Supplementary Fig. 7).

The differences in the prevalence of amino acid auxotrophies across different habitats matched differences in the taxonomic

**Table 1 | Attributes of the datasets included in the study**

| Habitat | #Samples | #Genomes | % ASVs with representative genomes | Proportional abundance of all ASVs with representative genomes (%) | % prototrophs | % with only 1 amino acid auxotrophy | Refs. |
|---|---|---|---|---|---|---|---|
| Rhizosphere | 230 | 2886 | 48.3 | 70.0 | 97.5 | 23.6 | 106 |
| Residential plumbing | 471 | 733 | 57.5 | 84.5 | 97.4 | 42.1 | 107 |
| Soil | 255 | 448 | 18.1 | 35.3 | 96.4 | 62.5 | 108 |
| Freshwater | 299 | 688 | 45.5 | 57.4 | 95.9 | 17.9 | 109 |
| Marine | 365 | 772 | 32.9 | 33.9 | 91.5 | 65.2 | 110 |
| Activated sludge | 514 | 1369 | 32.2 | 54.2 | 89.6 | 30.1 | 111 |
| Phyllosphere | 128 | 2949 | 43.6 | 41.9 | 89.4 | 9.9 | 112 |
| Cheese | 98 | 148 | 69.6 | 87.6 | 81.8 | 18.5 | 113 |
| Sourdough | 421 | 218 | 93.8 | 82.7 | 75.2 | 5.6 | 114 |
| Human skin | 335 | 1778 | 93.9 | 98.1 | 69.7 | 10.8 | 115 |
| Human gut | 350 | 979 | 66.3 | 89.1 | 41.6 | 25.7 | 116 |
| Human oral cavity | 347 | 571 | 83.7 | 97.9 | 24.7 | 9.1 | 115 |

Reference genomes were obtained by matching the 16 S rRNA gene amplicon sequences to the Genome Taxonomy Database (GTDB) allowing a single nucleotide mismatch. Genomes with a completeness lower than 95% were discarded. Only amplicon sequence variants (ASV) with more than 10 reads in a given habitat occurring in at least 10% of the samples were included. The proportion of single amino acid auxotrophs was calculated from the total number of auxotrophic taxa. The table is sorted by increasing proportion of amino acid auxotrophs in each habitat.

composition of the communities found in those habitats (Fig. 2B). Habitats dominated by the phylum Proteobacteria were the least auxotrophic, and habitats dominated by the Firmicutes were the most auxotrophic (Fig. 2B). These results agreed with the patterns we observed in the analysis across phyla (Fig. 1A), with families in the Firmicutes like *Lactobacillaceae* and *Streptococcaceae* being more auxotrophic than proteobacterial families like the *Pseudomonadaceae* or *Burkholderiaceae* (Supplementary Fig. 3). These results are unlikely to be biased by knowledge gaps in the amino acid biosynthesis pathways of the Firmicutes, as the Firmicutes is a well-studied phylum (see e.g. ref. [66]). Since we observed that assembled genomes had more auxotrophies than genomes from cultured isolates, we verified that the differences in the prevalence of auxotrophy across habitats were not driven by the proportion of assembled genomes and genomes derived from isolates across those habitats (Supplementary Fig. 8). Since the proportion of representative genomes recovered differed among habitats (Table 1), we also verified this proportion did not correlate with the proportion of auxotrophic taxa in those habitats ($r = -0.17$, $P = 0.6$).

As there are numerous examples of auxotrophic bacteria that have been isolated from soil[67,68], aquatic environments[1], food[24], plants[3], and the human gut[21,69,70], it has been assumed amino acid auxotrophy is a widespread trait across habitats. Our results indicate that amino acid auxotrophies are rather uncommon in non-host associated systems, and are only relatively common in host-associated systems (skin, gut, or oral cavity) and some fermented foods (cheese and sourdough) (Fig. 2A). The mean number of amino acids that bacterial taxa were unable to synthesize ranged between nearly zero in rhizosphere soils, residential plumbing, bulk soil, freshwater lakes, and marine surface waters, to 2–3 amino acids in taxa from the oral cavity, the human gut, and sourdough starter microbiomes (Fig. 2A). Host-associated habitats and fermented foods not only contained more auxotrophic taxa but those auxotrophs were unable to synthesize a larger number of amino acids (Table 1), suggesting that these environments generally support auxotrophic taxa[13]. Host-associated habitats often share a high and temporally stable supply of amino acids both from the host and ingested food[71], and fermented foods can have a high availability of peptides rich in amino acids such as milk proteins[72]. For example, in *Clostridium* species (phylum Firmicutes) amino acid auxotrophies have been associated with toxin production, which increases the availability of amino acids in the gut lumen[73]. We detected multiple amino acid auxotrophies in *Clostridium* species, which are capable of obtaining energy via the oxidation and reduction of amino acids using the

Stickland reaction in amino acid-rich environments[74]. Overall, our analyses suggest that amino acid auxotrophy might be most beneficial under conditions of temporally stable and (mostly) abundant amino acid supply, conditions which are not likely to be common in soils and aquatic environments. However, there are notable exceptions in these non-host associated environments. For example, while soils generally select for prototrophic bacteria (96.4% of soil taxa in our analyses were prototrophic, Table 1)[75], the common soil bacterium *Candidatus* Udaeobacter has a 'streamlined' genome with multiple amino acid auxotrophies that make it unique among soil bacterial taxa[76]. *Candidatus* Udaeobacter is considered a nutrient scavenger that likely benefits from the locally abundant nutrients provided by decaying bacterial biomass[76,77] (Supplementary Fig. 9). As another example, we found amino acid auxotrophies to be widespread among soil-dwelling Bdellovibrionaceae (Supplementary Fig. 9) and the predatory lifestyles of members of this group may allow amino acids to be obtained from ingested prey[78,79]. *Pelagibacter ubique*, an abundant pelagic bacterium with a highly streamlined genome[80,81]2, is another example of an organism with a free-living lifestyle where auxotrophy (in this case glycine auxotrophy[1]) is a successful strategy owing to the local abundance of glycolate (a precursor of glycine) from neighboring phytoplankton[82].

**Signatures of genome streamlining in amino acid auxotrophs**

As noted above, we found that auxotrophic taxa tend to have smaller genomes than prototrophic taxa and genome size was negatively correlated with the number of amino acid auxotrophies per genome (Supplementary Fig. 4C). This pattern is, in part, a product of obligate intracellular parasites having smaller genomes as a product of genetic drift[83], as would be the case for Spirochaetes and Tenericutes (Fig. 1A; Supplementary Fig. 3). However, this pattern could also be driven by auxotrophic free-living bacteria being more likely to have 'streamlined' genomes[84]. In other words, there is selection for amino acid auxotrophy in free-living taxa with smaller genomes that minimize cell complexity to more efficiently use the resources required to sustain growth. To test this 'streamlining' hypothesis, we focused our analyses on two phyla, Bacteroidetes and Firmicutes, with high proportions of auxotrophic taxa (37.2% and 37.0%, respectively), and we identified gene categories (COG categories[85]) that were differentially abundant across auxotrophic versus prototrophic members of each phylum (Fig. 3). In this analysis, we considered auxotrophic taxa to be only those taxa that were unable to synthesize two or more amino acids. In both phyla, genome size was negatively correlated with the number of

**A**

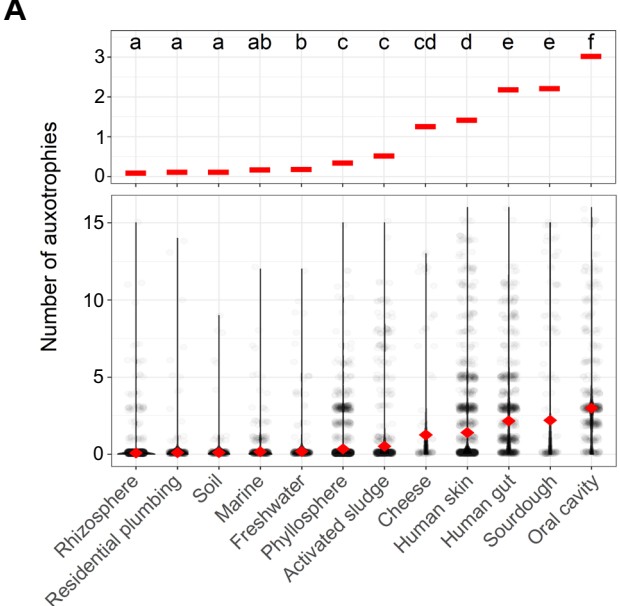

**B**

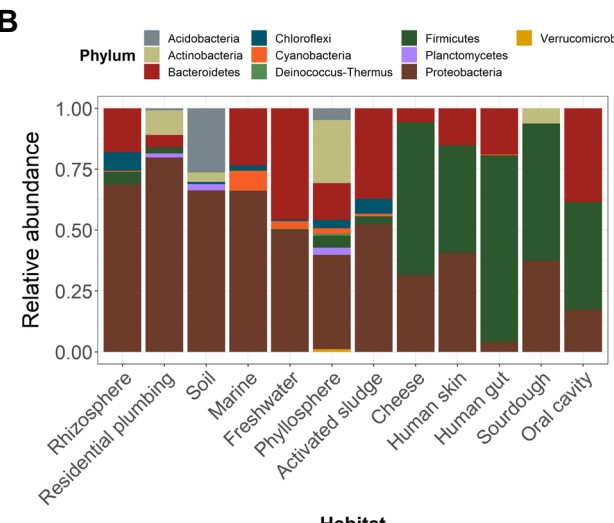

**Fig. 2 | Amino acid auxotrophy in bacteria across habitats. A** Prevalence of amino acid auxotrophy in representative bacterial taxa from 12 different habitats (*N* = 13,523 genomes). The mean number of amino acid auxotrophies of representative bacterial taxa in each habitat is shown as red diamonds in the main panel, and as horizontal bars in the top subpanel. **B** Relative abundance of the most dominant phyla across habitats. The x-axis is sorted by increasing numbers of auxotrophic taxa in each habitat. Letters at the top of **A** indicate statistical differences (*P* < 0.05) between habitats based on Mann-Whitney U tests with Bonferroni-corrected *p*-values.

amino acid auxotrophies per genome (Fig. 3A, B), in agreement with the general expectation from streamlining theory[84]. Likewise, as expected for streamlined taxa, genes for translation, protein turnover, and post-translational modification were all overrepresented in the genomes of auxotrophic taxa (Fig. 3C). These and other functional gene categories, such as nucleotide transport and metabolism and DNA replication, recombination and repair have all been previously linked to genome streamlining and associated life history strategies across a broad range of bacterial taxa[75,86,87]. The genes overrepresented in the genomes of prototrophic taxa were also consistent with our expectations and previous findings: genes for the transport and metabolism of carbohydrates, amino acids, and lipids, and genes for

transcription and signal transduction were all overrepresented in the genomes of prototrophic taxa (Fig. 3C)[86,87]. Together, these findings indicate that amino acid auxotrophy is part of the general life history strategy that characterizes bacteria with 'streamlined' genomes.

## Conclusions

Amino acid auxotrophy is broadly distributed across the bacterial tree of life, but it is likely less common than previously assumed. We observed appreciable taxon-specific and habitat-specific differences in the prevalence of amino acid auxotrophies, whereby amino acid auxotrophy seems to be most prevalent in host-associated systems or habitats where amino acid availability is expected to be relatively high. In free-living taxa, amino acid auxotrophy likely arises as a product of the genome streamlining process, whereby taxa are adapted for efficient growth sustained on temporally stable supplies of nutrients. This strategy is likely a characteristic of the majority of bacterial taxa that remain uncultured[42], emphasizing the need for directing culturing efforts towards bacteria with traits such as auxotrophy and small genomes. Overall, our comprehensive investigation of bacterial amino acid auxotrophies highlights that we still have insufficient experimental evidence to confirm amino acid auxotrophies across many bacterial groups. Dedicated culturing and testing of growth requirements across diverse bacterial taxa would further our understanding of the links between auxotrophy and the specific bacterial life history strategies that make amino acid auxotrophy an ecologically successful strategy.

## Methods

### Study design

We compiled the full sequences of the ~62,000 unique bacterial genomes ('species clusters') available in the Genome Taxonomy Database (GTDB) (release 207)[53]. We restricted our analyses to only those bacterial phyla with more than 100 representative genomes available in GTDB (12 phyla in total) and only included genomes estimated to be >95% complete based on CheckM (v1.1.6)[88]. We also removed all metagenome-assembled genomes (MAGs) that lacked 16 S rRNA genes, as well as those with signals of chimerism based on GUNC (Genome Unclutterer)[89], yielding 26,277 genomes in total. We then ran the automated amino acid biosynthesis annotation tool GapMind on all of these genomes[31]. GapMind identifies candidates for steps in amino acid biosynthesis by using a database of 1849 proteins that have been experimentally shown to be involved in amino acid biosynthesis (taken primarily from MetaCyc[90], SwissProt[91] and BRENDA[92]), as well as 145 protein families (144 TIGRfams[93] and 1 Pfam[94]). GapMind then searches genomes for candidates in the reference biosynthesis pathways using ublast (for similar proteins[95]) or HMMER (for members of the same protein family[96]), providing confidence of matches based on sequence identity and coverage[31]. At this step, GapMind uses ublast to check if these candidates are similar to any of 113,704 experimentally-characterized proteins that could have alternative functions to amino acid biosynthesis. Candidates are considered valid if the bit score of the alignment to proteins involved in amino acid biosynthesis is higher than the bit score of the alignment to proteins with other functions. We considered a biosynthetic step to be present if it had at least a medium-confidence candidate, which for protein candidates based on similarity to a characterized protein means either (1) at least 40% identity and 70% coverage to a characterized protein, or (2) at least 30% identity and 80% coverage and more similar to protein(s) with this function than to another characterized protein in the database of the 113,704 proteins. We predicted the biosynthesis capabilities for 17 amino acids and chorismate (a precursor of aromatic amino acids), but excluded alanine, aspartate, and glutamate because these amino acids can be produced via the transamination of intermediates from central metabolism, and annotating the substrates of transaminases is inherently challenging[29].

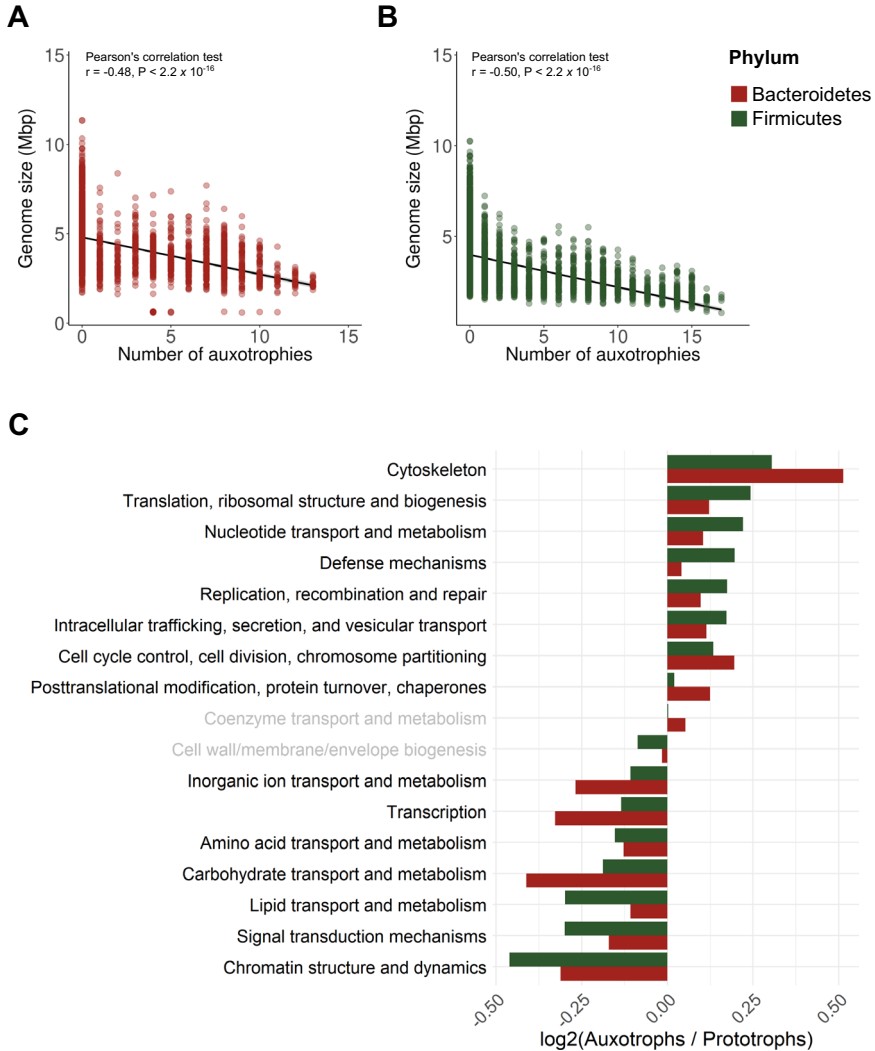

**Fig. 3 | Evidence for genome streamlining and related functional features in bacterial taxa that are auxotrophic for amino acids. A, B** Relationship between genome size and the number of amino acid auxotrophies in bacterial taxa from the phylum Bacteroidetes **A** and Firmicutes **B**, respectively. **C** Functional categories that are consistently overrepresented in auxotrophic and prototrophic genomes from the phyla Bacteroidetes and Firmicutes. Functional categories were defined as Clusters of Orthologous Genes (COGs). We displayed those categories where the Bacteroidetes and Firmicutes had non-statistically significant trends in grey font based on Mann-Whitney U tests ($P < 0.01$). Pearson's correlation coefficients ($r$) are displayed on **A, B**. $N_{Bacteroidetes}$ = 3232 genomes, $N_{Firmicutes}$ = 4674 genomes.

In addition to predicting amino acid auxotrophy across bacterial phyla, we also investigated how the prevalence of amino acid auxotrophy varies across different bacterial habitats. To do so, we used 16 S rRNA gene sequencing data from 12 different habitats (one dataset per habitat, Table 1), to identify the predominant bacterial taxa found in each of the 12 habitats. We selected 12 publicly available 16 S rRNA gene sequence datasets that each had >100 samples, with each dataset including a broad range of sample types representative of the habitat. These datasets were analyzed using the same bioinformatic pipeline. Briefly, we used *cutadapt* (v1.18)[97] to remove primers, adapters and ambiguous bases from the 16 S rRNA gene reads. We then quality-filtered the sequences, inferred amplicon sequence variants (ASVs) using the DADA2 pipeline (v1.14.1)[98], and removed chimeric sequences. Taxonomic affiliations were determined against the SILVA SSU database (release 138)[99]. We used the *phyloseq* R package (v1.38.0)[100] for downstream analyses. From each dataset we obtained representative genomes by matching the 16 S rRNA gene sequences of individual taxa to genomes in GTDB, allowing a single base mismatch (i.e. 99.6% sequence similarity for 250 bp fragments), following the approach used previously to investigate the genomic attributes of bacteria across environmental gradients[101]. We only included ASVs that had

more than 10 reads in a given habitat and occurred in at least 10% of the samples from each dataset as we wanted to focus on representative genomes from those taxa that are reasonably ubiquitous in each of the 12 habitats. We ran the GapMind pipeline on these representative genomes to infer the completeness of the amino acid biosynthesis profiles for those bacterial community members in each habitat.

**Validation of amino acid auxotrophy predictions**
Since many of the genes involved in amino acid biosynthesis are not well described[29], genome-based inferences can significantly overestimate the prevalence of auxotrophies. Thus, to validate our approach, we compiled genomic information from 171 taxa that are known to grow in minimal media without the external supply of amino acids (i.e. prototrophs, compiled in Price et al.[31]; Supplementary Data 1) and ran GapMind on those genomes to quantify biases in our predictions. We also estimated the accuracy of the predictions for specific auxotrophies by compiling genomic information for 19 taxa with experimentally determined auxotrophies (compiled from[31,102]; Supplementary Table 1). This validation allowed us to determine the number of genes that need to be missing in any given amino acid biosynthesis pathway to consider that taxon auxotrophic for a given amino acid.

## Associations between functional genes and amino acid auxotrophy

We investigated associations between amino acid auxotrophy and broad functional gene categories by testing the prevalence of Clusters of Orthologous Genes (COGs) in the genomes of auxotrophic and prototrophic taxa[85]. We conducted these analyses on the phyla Bacteroidetes and Firmicutes as the metabolic pathways of these phyla are relatively well-studied, contain >3000 taxa with available genomes, and these phyla include sizeable proportions of auxotrophs for robust statistical analyses. We annotated genomes into COG categories using eggNOG-mapper v2[103], and calculated the genome size-corrected prevalence of each COG category per genome. In order to have a conservative classification of auxotrophy, we only classified those taxa that contained 2 or more amino acid auxotrophies as auxotrophs, and those taxa with no auxotrophies as prototrophs. We obtained minimal doubling times for all genomes based on the predictions established by Weissman et al.[104] (gRodon R package; https://github.com/jlw-ecoevo/gRodon), by matching the genome accessions of the taxa in the EGGO database (https://github.com/jlw-ecoevo/eggo).

## Statistical analyses

We verified the non-normality of the data using the Shapiro-Wilk test and compared the number of auxotrophic taxa between phyla and habitats using Mann-Whitney U tests using the wilcox.test() R function with Bonferroni correction of $p$-values for multiple comparisons. We used Pearson's correlation tests to determine whether bacteria were more auxotrophic for amino acids with higher biosynthetic energy costs. The same test was used to investigate correlations of auxotrophy with genome size. We obtained information on the energy (P-bonds) required for amino acid biosynthesis from Akashi and Gojobori[12]. We used multiple Mann-Whitney U tests with Bonferroni correction for multiple comparisons to investigate whether particular COG categories were overrepresented in genomes from auxotrophic versus prototrophic taxa. We represented the results as the log2-fold ratio. Finally, we investigated associations between the estimated bacterial minimal doubling times and genome origin using Mann-Whitney U tests, and tested differences in genome size between assembled genomes and genomes from cultured isolates using Welch two-sample two-sided t-tests. All statistical analyses were performed in R (v4.1.3)[105].

## Reporting summary

Further information on research design is available in the Nature Portfolio Reporting Summary linked to this article.

## Data availability

All sequence data analyzed for this study had already been deposited in open repositories and can be accessed through the specific works cited in this work. The source data to reproduce the findings of this study has been made publicly available on Figshare (https://doi.org/10.6084/m9.figshare.24101742.v1). The genome data included in this study can be found in the Genome Taxonomy Database (GTDB, https://data.gtdb.ecogenomic.org/releases/release207/207.0/). Information on predicted doubling times in bacteria can be found in the EGGO database (https://github.com/jlw-ecoevo/eggo). Functional gene annotations were based on the Database of Clusters of Orthologous Genes (COGs, https://www.ncbi.nlm.nih.gov/research/cog).

## Code availability

The code to reproduce the findings of this study has been made publicly available on Figshare (https://doi.org/10.6084/m9.figshare.24101742.v1).

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

## Acknowledgements

We thank all the people involved in the acquisition of the data compiled in this study. J.R. acknowledges funding from the Swiss National Science Foundation (Early PostDoc Mobility grant P2EZP3_199849). N.F. was supported by grants from the National Science Foundation (OPP 2133684 and AW5809-826664). M.N.P. was funded by ENIGMA—Ecosystems and Networks Integrated with Genes and Molecular Assemblies (http://enigma.lbl.gov), a Science Focus Area Program at Lawrence Berkeley National Laboratory is based upon work supported by the U.S. Department of Energy, Office of Science, Office of Biological & Environmental Research under contract number DE-AC02-05CH11231. E.O.C. was supported by grants from MICIN/AEI/ERDF (INTERACTOMA RTI2018-101205-B-I00).

## Author contributions

J.R. and N.F. conceived and designed the study. J.R. and T.B.N.J. performed the data analysis with the help of M.N.P. E.O.C. and M.N.P. provided data to the study. J.R. and N.F. wrote the manuscript, with input from all co-authors.

## Competing interests

The authors declare no conflicts of interest.
