## [Peer Review File NEW · Nature Communications]

Taxonomic and environmental distribution of bacterial amino acid auxotrophiesReviewer #1 (Remarks to the Author):

The manuscript "An ecological perspective on bacterial amino acid auxotrophy" by Ramoneda et al. submitted for publication in Nature Communications provides a comprehensive analysis of bacterial auxotrophy and estimates how prevalent amino acid auxotrophy is across different habitats. The authors describe the results from the utilization of a bioinformatic tool called GapMind, which had been previously published, to 26,277 genomes and MAGs/SAGs derived from ~62,000 genomes available at GTDB to investigate the proportion of amino acid auxotrophy. Subsequently, the authors describe the utilization of the same software to determine the prevalence of amino acid autotrophy across different samples from 12 different ecological habitats, and investigate auxotrophy at different taxonomic levels. Finally, the authors describe the statistical relationship of the number of identified auxotrophies with doubling time, genome size, GC content, and energetic costs of the biosynthesis of amino acids.

MAJOR COMMENTS

Understanding auxotrophies is an important topic in microbiology. Unfortunately, the work by Ramoneda et al does not provide any new insight into the topic. The authors use an existing tool (GapMind) and apply it to a large set of genomes to provide, as the title states correctly, an estimate of auxotrophies. There are several shortcomings that should have been addressed. The authors use an arbitrary cutoff of 40% (genes missing) to benchmark the "accuracy" of their tool on bacteria from 4 phyla, heavily biased against Bacillota (Firmicutes). Based on this, they assume that this cutoff can be applied to all phyla – but do not bother to confirm any of these predictions experimentally. Note that the authors used identical organisms before for the original GapMind publication and do not provide a justification for why new benchmarking is required here. The bias towards data validation from one phyla (Firmicutes) is even more peculiar, because predicted auxotrophies (Fig 2) matches pretty well the abundance of Firmicutes. Additionally, Firmicutes are also highly abundant in host-associated habitats - hinting that one of the main findings (host-associated habitats contain microbes with higher auxotrophies) could be due to this skewed analysis. To rule out any bias here, I would have expected substantial experimental validation, e.g. for some key bacteria mentioned in the study (members of the Lactobacillaceae and Streptococcaceae). Another finding that would require detailed validation would be for members of the Deinococcus-Thermus phylum. All 24 taxa within the Deinococcus-Thermus are supposed to be auxotroph for serine – this can easily be validated by testing for example 10 of those taxa (the taxa also has the highest percentage of isolate-derived genomes compared to MAGs/SAGs genomes). Why was this not done? Because of the type of analysis, the findings are very vague, do not provide any hard data, and result, as the title says, in an "estimation". However, the authors provide numbers with decimals, e.g. 99.2%, inferring a type of accuracy that is not given. This estimation is then linked to functional categories (Fig 3), to generate yet another correlation (based on very soft data to start with), using a second tool (eggNOG-mapper). Wouldn't this tool also be able to predict gene presence/absence calls for AA biosynthesis genes? Also, it is unclear if the statistical analyses used throughout the manuscript are appropriate for the type of data. Specifically, are Mann-Whitney U-test appropriate when analyzing stratified/discontinuous data (Fig 1A and B, Fig 2A) and Pearson's correlation appropriate to compare discontinuous to continuous data (Fig 3A and B) information?

The original GapMind study used 148 genomes of highest quality for benchmarking and conservative (but not perfect of course) usearch parameters for global alignment. The discussion section of the GapMind paper ends with the disclaimer: "However, we still do not understand how most bacteria or archaea can make all 20 amino acids". The current manuscript ignores this fact and provides an analysis that is based on too many assumptions, including sparse genomic data from MAGs and SAGS, to generate correlation-based results that are lacking validation.

MINOR COMMENTS:

L41/L43 The authors should clearly differentiate between bioinformatic results and experimental/previous results.

L67/L69 Does "outcompetition" refer to the growth rate of the auxotrophic strain in the original medium containing the amino acid?

L76 please rephrase - Here, we focus on amino acids

L77 please rephrase - ...amino acids can be important both

L91 "Auxotrophy is expected to be more common ..." as the sentence is framed as speculative.

L92 Does the 20-30% refer mainly to amino acids in proteins and/or small (poly)peptides? Also, is the term "dissolved" used to describe free amino acids? Are these single AAs or include peptides?

L93 "N" was never introduced as an acronym for "nitrogen", and the Methods section use "N" for the sample size. (Also, inconsistent use of "P" and "p" for p-value)

L100 because of their spatial proximity?

L104ff the authors indicated above that their focus is on amino acid auxotrophies, but mix AA auxotrophies and other auxotrophies in the introduction text. It would help to be more explicit here, maybe use the term AA-auxotrophies to clearly differentiate the different types.

L152 what are those 171 taxa? Is there a list that shows the phylogenetic representation of the taxa (see comment below for L159)

L159 of course – you used a manually created cutoff (40%) to achieve this accuracy..., while this is a valid method to benchmark your approach, it is still arbitrary and could be biased (based on phylogeny, genome size and quality). The validation is performed on only 19 taxa. These 4 phyla are heavily biased towards Bacillota, aka Firmicutes (11/19 of the members belong to this phyla). The other members are 5 Proteobacteria, 2 Fusobacteriota and 1 Verrucomicrobiota. It is also curious why *E. coli*, *B. subtilis* or other model organisms are missing – are there no experimental validation for those microbes?

L203 one study – two references?

L205 unfortunately – after all... it is just an "estimate", and the authors argue that their estimate is superior over previous estimates (using an arbitrary cutoff of 40% genes missing)

L222 ...group, which is...(comma)

L235 why are bacterial families referenced here? What information from the references is being used in the discussion of the results?

L242 given the difference in auxotrophies from MAGs/SAGs and isolate genomes, I would strongly argue against the use of any MAGs/SAGs in the analysis! SAGs in particular are of low completeness (and SAGs and MAGs from the same samples seldomly match...)

L247 "higher minimal doubling times" for uncultivated taxa? Were these doubling times predicted? How? Approaches based on replication rates (e.g. Brown et al Nat Biotech 34, 2016) have been proven to be highly variable and not providing consistent data across different samples, containing different taxa

L248 cultured isolates can be from different media/culture conditions. The result of a statistical test is invalid if differences in culture conditions were not controlled

L251 a (negative) significant relationship with GC might be anecdotal rather than a causation. Gene duplication and long intergenic regions can significantly modify the GC content of a genome

L253 The authors state that observed auxotrophies in MAGs/SAGs are likely due to genome size, a potentially misleading statement that the genome size is a cause of auxotrophy rather than a (weak) correlation. Also, the method to determine the correlation seems not appropriate for discrete data (number of auxotrophies) versus continuous data (genome size)

L255 what does "complete" mean in this context? More prototrophic?

L255 completeness of MAGs is also not "exact science" but based e.g. on marker genes presence/absence (see e.g. CheckM). Completeness of MAGs has recently been linked to the recovery of functional signal (10.1038/s43705-023-00221-z), so I would argue that auxotrophy assignment is indeed linked to completeness bias

L258 Extended Data Fig 4 is not very meaningful (i.e. percent). Representation of absolute numbers would be better suited here

L261 general statements including entire taxa are highly speculative and thus not very meaningful

L264 "larger genomes compared": Average size of genomes might also be biased towards genomes with gene copies of unrelated pathways

L270ff is this maybe because some biosynthesis pathways require less genes and thus the 40% cutoff is heavily biased? L286 shows a test for tryptophan (with a not very strong r and p value), why was this not done for all AAs?

L284 "unsurprisingly" – please rephrase, it looks like the authors refer to gut-associated bacteria rather than to the specific genera

L286/L287 A correlation between number of genes in a certain pathway and the number of auxotrophies is unnecessary and derives from an arbitrary definition of pathway

L289 "a previous work" or "previous works"?

L289/L294 The authors should mention the coefficient value of the correlations found in previous works. Also, the correlation might be misleading, as the correlation can be associated also to the total number and type of bacteria analyzed previously

L309 Fig 2 shows that auxotrophy matches abundance of Firmicutes – to rule out any bias, I would expect experimental validation of some key bacteria for the Lactobacillaceae and Streptococcaceae. Also, see L159, the algorithm was benchmarked on data that contained ~60% Firmicutes.... Coincidence?

L316 Extended Data Fig 6 – why only three specific proteobacteria (alpha, beta, gamma)? Why not all proteobacteria? Or is there supposed to be a separation between the different proteobacteria? The graph needs a suitable legend to explain what is shown (i.e. statistics, lines, dots).

L329 How can this be shown by looking at percentages? Wouldn't this have to be checked for what organisms are preferably from MAGs/SAGs? For example, if all Firmicute data comes from MAGs/SAGs, the overall percentage shown in Extended Data Fig 8 would not be suitable to make this statement. In the next sentence the authors try to address this (reporting a poor r and p value) – indicating that in fact the analysis is biased toward the proportion of representative genomes.

L329 Extended Data Fig 8 comes before Extended Data Fig 7 in the text

L352 It was stated in the introduction that available proteins do not necessarily translate into available/dissolved amino acids. How is this relevant in this context, but not before?

L354 what is "stable"? These statements are way to general to be of any meaning to the reader (especially since you are referring to very broad habitat classifications)

L356/L373ff instead of providing yet another genome-centric analysis, why not confirm some of these predictions?

L390ff ...which has been shown before as you correctly state. The entire paragraph does not provide any new insights but reiterates current knowledge. I am not sure what is gained here.

Methods:

L451 Is "medium-confidence" classification provided by the software, or did the authors use a threshold value to filter results from GapMind? Also, can you please provide the computer characteristics and typical run time per genome?

L453 The second criterion "at least 30% identity and 80% coverage and more similar to proteins with this function than to any other characterized protein in a database of >100,000 characterized proteins" lacks details such as the source of this database (GapMind uses 6,284 experimentally characterized proteins) and what is "similarity"/"more similar" as the authors use identity and coverage.

L483/L495 Why is validation of the predictions important? As far as I understand, the authors used published software... It is not clear what is different from the previous validation, or if this new validation step introduced biases in the analysis rather than quantifying them.

L497/L512 This section describes the use of eggNOG-mapper to (re-)annotate the genomes and obtain COG classification, and normalize the prevalence to the genome sizes. However, it is not clear why the authors did not compare results from eggNOG-mapper and GapMind. Also, it describes the use of doubling times, but it is not clear if the authors used the gRodon software, or they used data from the EGGO database. If gRodon, the authors should describe briefly the procedure to run the software. If EGGO, the authors should describe how they matched genomes from GTDB to the data available in EGGO. Finally, it is not clear why >3000 taxa per phyla are necessary for the statistical tests.

References:

Please assure to italicize species names and not to mix different styles (e.g. all first letters capitalized)

Figures:

Figure 1 Style between Fig 1A and 1B is different. Specifically, 1B hides the stratification of the data. It is suggested that Fig 1B style be identical to Fig 1A. The scale between Fig 1A and 1B is different. Also, Fig 1A shows labels on top (a, b, etc) that are not explained. The numbers next to the taxon names in Fig 1C lack explanation as well. Why are they different from Fig 1A?

Figure 2 Top panel of Fig 2A is redundant with red diamonds in the bottom panel. Also, letters referring to "statistical differences" don't show what pairwise samples were compared to obtain these differences. In Fig 2B, some ASVs cannot be assigned to a phylum. Is the proportion of unassigned ASV low or zero? Can the authors provide their abundance? Also, 10 phyla were identified, while in Fig 1A and C, 12 phyla were analyzed. Was the abundance of the missing 2 phyla too low?

Figure 3 Significant level was defined as p-value less than 0.01, that is an atypical value. No explanation of the defined p-value was provided, as it is expected that lowering the threshold for the significance, it will reduce the number of categories that are significantly different.

Reviewer #2 (Remarks to the Author):

The authors present a different approach and perspective on an important topic in microbial ecology. As the authors lay out in the manuscript there are many gaps in our understanding of the prevalence of bacterial auxotrophy and it is challenging to use genomic approaches to confidently identify the absence of pathways. I found the authors' conservative approach to predicting amino

acid auxotrophy and the manner in which they validated their results very thorough. While their approach identified fewer bacterial auxotrophs than other approaches, many of the trends in taxa and habitat where auxotrophs are most prevalent were consistent with previous work. I have only a couple minor comments and questions below. The paper was clear and well written.

L160: Do the authors have any ideas as to why their model did not perform quite as well with serine and cysteine? Are there more possible synthetic pathways for arriving at these particular amino acids? It seems helpful to discuss the reasons for this to help think about how to approach predicting auxotrophy for other metabolites beyond amino acids.

L500-502: I'd suggest mentioning that you required 2 auxotrophies to count a taxa as an auxotroph in the results & discussion. I think readers will assume that a single auxotrophy would be counted as an auxotroph and might miss this information in the methods section. I also think it would be worth mentioning in the methods or extended data how many more auxotrophs you would have if you did not require 2 auxotrophies. I'm curious how much this changes the predictions. I understand that the authors were trying to be conservative in the identification of auxotrophs, but this seems like it could change the results quite a bit. But, maybe it doesn't. Are most experimentally identified auxotrophs auxotrophic for more than one amino acid?

Reviewer #3 (Remarks to the Author):

Ramonedá et al. present a well-written analysis of the prevalence of amino acid auxotrophy across 12 bacterial phyla and study auxotrophy patterns across environments using a validated model to improve on previous estimations of amino acid auxotrophy in bacteria.

I recommend this manuscript for publication with one major and a few minor revisions.

MAJOR

The authors use an arbitrary 40% cutoff for the auxotrophy designation throughout their manuscript despite the fact that their data show this cutoff is reliable for some amino acids and too high for others such as serine and cysteine, which they acknowledge (see Extended data figures 1A and 1B, which show the number of false positives). This is a missed opportunity to use the information they have already presented to determine a tailored cutoff for each amino acid to improve accuracy. Doing so is especially important given that the authors use their data to estimate the percentage of all (sequenced) bacteria that are prototrophs.

MINOR

General:

The words "Ecological perspective" in the title were misleading, I thought the manuscript was going to be a hypothesis piece rather than a research article. Should be retitled to capture everything that they did more specifically.

The authors allude to the cultivability bias in their dataset, but it should be acknowledged explicitly.

Figure 1A and 2A captions: Explain what the diamonds are

The authors should explain why only 17 amino acids were chosen and why chorismate was included in the analysis.

Extended data figures 7 and 8 are not mentioned in order in the text.

Specific:

91-96 should include numbers for total N in addition to amino acid percentages since availability will depend on total amino acids. Lines 353-356 contradict the statement about marine and freshwater environments.

103 shorten sentence to improve readability

127 contains a typo "are remain"

199 The definition of taxa here is unclear, is this referring to individual genomes?

219 It is unclear what the expectation was for Cyanobacteria

245 what proportion of MAGs / SAGs were from uncultivated taxa?

250 The authors explain the relationship between auxotrophy and genome size but do not explain the GC content relationship.

254 It is unclear what the percentages in parentheses are for

345 "Select for" is too strong, please rephrase it to something like "Support"

377 typo: should refer to fig 3C, not 4C

435 How was genome completeness estimated?

See below the responses to the reviewer comments on manuscript NCOMMS-23-30557-T, which we have thoroughly addressed. We would like to thank the three independent reviewers for their constructive feedback, which we found very useful for improving the content, clarity, and quality of this work.

Reviewer #1:

The manuscript “An ecological perspective on bacterial amino acid auxotrophy” by Ramoneda et al. submitted for publication in Nature Communications provides a comprehensive analysis of bacterial auxotrophy and estimates how prevalent amino acid auxotrophy is across different habitats. The authors describe the results from the utilization of a bioinformatic tool called GapMind, which had been previously published, to 26,277 genomes and MAGs/SAGs derived from ~62,000 genomes available at GTDB to investigate the proportion of amino acid auxotrophy. Subsequently, the authors describe the utilization of the same software to determine the prevalence of amino acid autotrophy across different samples from 12 different ecological habitats, and investigate auxotrophy at different taxonomic levels. Finally, the authors describe the statistical relationship of the number of identified auxotrophies with doubling time, genome size, GC content, and energetic costs of the biosynthesis of amino acids.

MAJOR COMMENTS

Understanding auxotrophies is an important topic in microbiology. Unfortunately, the work by Ramoneda et al does not provide any new insight into the topic. The authors use an existing tool (GapMind) and apply it to a large set of genomes to provide, as the title states correctly, an estimate of auxotrophies.

There are several reasons why this work provides new insights into the topic of microbial auxotrophy: 1) Our study covers >26,000 genomes across 12 phyla, an appreciably larger number than the previous effort which included amino acid auxotrophy predictions for 949 genomes. 2) Predicting auxotrophy is a key novel aspect of the work. GapMind annotates pathways, but it does not predict auxotrophies. To predict auxotrophies, we introduced a 40% threshold and validated its performance in distinguishing prototrophic and auxotrophic bacteria. 3) We provide detailed information on the overall prevalence of amino acid auxotrophies and how the prevalence of such auxotrophies varies across major microbial habitats, demonstrating that some environments harbor far higher proportions of auxotrophic bacteria than others. 4) We conducted a detailed and comprehensive analysis of the general life history strategies associated with amino acid auxotrophy which, to our knowledge, is novel as these had remained speculative in the past.

There are several shortcomings that should have been addressed. The authors use an arbitrary cutoff of 40% (genes missing) to benchmark the “accuracy” of their tool on bacteria from 4 phyla, heavily biased against Bacillota (Firmicutes). Based on this, they assume that this cutoff can be applied to all phyla – but do not bother to confirm any of these predictions experimentally. Note that the authors used identical organisms before for the original GapMind publication and do not provide a justification for why new benchmarking is required here.

Thanks for raising this point. To address these issues, we now provide updated Extended Data Figures 1 and 2 where we show how our predictions on the validation dataset would change if we were to use different cutoff values (% of genes missing) to infer auxotrophy. Finding a minimum number of genes that needed to be missing in a given amino acid biosynthesis pathway was not a way to benchmark accuracy, but a way to turn genome annotations into predictions of auxotrophy. We conservatively chose a cutoff that minimizes the incorrect prediction of auxotrophy as gaps in our knowledge of specific amino acid biosynthesis genes can lead to the overestimation of auxotrophy. Note that the validation to obtain this 40% 'missing genes cutoff' was based on information from 12 phyla, not 4 as mentioned here. The reviewer is correct that only 9 out of 171 prototrophic taxa used for validation are from the phylum Firmicutes – however, this does not necessarily mean the likelihood of overestimating auxotrophy in this group is higher. The Firmicutes are a relatively well-characterized phylum, so it is unlikely our general patterns are biased by knowledge gaps in this group or insufficient empirical information. Generally, new benchmarking (i.e. experimental determination of amino acid auxotrophy across a broader diversity of taxa) would be required to improve annotation tools and this is a point we highlight in the Discussion. However, we are confident from our analyses of the validation datasets (again, see Extended Data Figures 1 and 2) that the 40% 'cutoff' chosen for this study is appropriate and provides conservative inferences of the prevalence of amino acid auxotrophies across a broad range of taxa and habitats.

The bias towards data validation from one phyla (Firmicutes) is even more peculiar, because predicted auxotrophies (Fig 2) matches pretty well the abundance of Firmicutes. Additionally, Firmicutes are also highly abundant in host-associated habitats - hinting that one of the main findings (host-associated habitats contain microbes with higher auxotrophies) could be due to this skewed analysis.

As mentioned above, the determination of the missing genes cutoff was based on 171 empirically validated amino acid prototrophs, not on the 19 auxotrophs (of which indeed 11 are Firmicutes). There is no connection between the overestimation of auxotrophy in a given group and how well-represented this group was in the validation dataset (as the name indicates, it is data for validation, not a training dataset to build any model). The factors that can skew the predictions are related to existing knowledge on the enzymes involved in amino acid biosynthesis in bacteria, which we now explicitly acknowledge for cysteine and serine, where GapMind had difficulties in finding candidate genes. We found that all wrongly predicted cysteine and serine auxotrophs are bacteria with sulfur-related metabolisms (Desulfobacteria and Aquificae), which can synthesize serine from pyruvate, but the genes involved are not known. Crucially, these groups were not included in our main analyses, so the likelihood that our analyses are biased towards particular groups is small. We have revised the text to highlight these points and included supporting references (lines 159-174).

To rule out any bias here, I would have expected substantial experimental validation, e.g. for some key bacteria mentioned in the study (members of the Lactobacillaceae and Streptococcaceae).

Thank you. As mentioned in the text, the goal of this study was not to improve current amino acid biosynthesis annotation tools, but to turn the gene annotations derived from a well-established and validated tool (GapMind) into accurate predictions of auxotrophy. We recognize (and highlight on lines 206-208) that additional experimental studies that couple extensive media testing, whole genome sequencing, and comparative genomics are needed to yield new insights into bacterial amino acid metabolism (work such as that presented in this study which set the basis for the development of GapMind -

<https://journals.plos.org/plosgenetics/article?id=10.1371/journal.pgen.1007147>). Finally, the key bacteria mentioned (Lactobacillaceae and Streptococcaceae) are groups for which there is already ample experimental information on auxotrophy because these are frequently isolated and characterized (for reference, see e.g. <https://journals.asm.org/doi/10.1128/aem.70.3.1869-1873.2004> or <https://journals.asm.org/doi/10.1128/jb.183.24.7354-7364.2001> and citations within). This means additional experimental validation on these groups is not likely to yield much additional information to validate the predictions of this study. Rather, the fact that our predictions are in accordance with the ample information available for these groups suggests that our predictions are robust.

Another finding that would require detailed validation would be for members of the Deinococcus-Thermus phylum. All 24 taxa within the Deinococcus-Thermus are supposed to be auxotroph for serine – this can easily be validated by testing for example 10 of those taxa (the taxa also has the highest percentage of isolate-derived genomes compared to MAGs/SAGs genomes). Why was this not done?

This is a great point. Indeed, a novel type of phosphoserine phosphatase has been discovered in *Thermus thermophilus* (<https://doi.org/10.1111/febs.14703>) and this enzyme was not included in the GapMind framework. We now acknowledge that this is likely the reason for the misclassified serine auxotrophies in the Deinococcus-Thermus phylum. See lines 311-314.

Because of the type of analysis, the findings are very vague, do not provide any hard data, and result, as the title says, in an “estimation”. However, the authors provide numbers with decimals, e.g. 99.2%, inferring a type of accuracy that is not given.

We are not quite sure how to address this point as we strongly disagree that our findings are ‘vague’, nor do we agree that we ‘do not provide any hard data’. We analyzed over 26,000 genomes for the genomic analyses and we used data compiled from 3,813 samples spanning major aquatic, terrestrial, and engineered environments to determine habitat-specific and taxon-specific differences in the prevalence of amino acid biosynthetic capabilities. Given the enormous amounts of data that went into this study, we see no reason to round the proportions of taxa that were auxotrophic by phylum or habitat, as the numbers were obtained from large sample sizes. We do agree that the decimals in our accuracy estimates are misleading, so we have rounded them in the revised version and appreciate the suggestion.

This estimation is then linked to functional categories (Fig 3), to generate yet another correlation (based on very soft data to start with), using a second tool (eggNOG-mapper). Wouldn't this tool also be able to predict gene presence/absence calls for AA biosynthesis genes?

The functional categories mentioned here are used to represent life history strategies, which by definition are correlations of multiple traits, so there is no reason to think our correlation analysis is problematic. We do not know what the reviewer means by ‘very soft data’ – these types of analyses are widely used in many areas of biology and microbiology. Regarding the use of eggNOG-mapper, please note that this framework is not suitable for the annotation of amino acid biosynthesis pathways as its annotation pipeline is limited to databases that classify proteins based on broad functions and protein families (such as COG or Pfam). GapMind's annotation database is tailored to specifically identify genes involved in amino acid biosynthesis, and that is why it is the most appropriate tool for this study. We have provided more details on how GapMind works to emphasize this point (see L472-482).

Also, it is unclear if the statistical analyses used throughout the manuscript are appropriate for the type of data. Specifically, are Mann-Whitney U-test appropriate when analyzing stratified/discontinuous data (Fig 1A and B, Fig 2A) and Pearson's correlation appropriate to compare discontinuous to continuous data (Fig 3A and B) information?

Thanks for raising this point. The documentation of the R function `wilcox.test()` we used to perform the Mann-Whitney test explicitly highlights that such tests are appropriate for stratified data (this is, the inclusion of ties). Pearson's correlation is an appropriate test as long as the data is quantitative and the groups are independent, and it does not make any assumptions about the continuity of the data. Both methods are therefore appropriate for the analyses presented here. We now specify that Mann Whitney U tests were performed using the `wilcox.test()` R function in the methods section.

The original GapMind study used 148 genomes of highest quality for benchmarking and conservative (but not perfect of course) usearch parameters for global alignment. The discussion section of the GapMind paper ends with the disclaimer: "However, we still do not understand how most bacteria or archaea can make all 20 amino acids". The current manuscript ignores this fact

It's true that for many bacteria, we don't fully understand how they make all 20 amino acids. But mostly, these are cases where a well-established pathway seems to be present, but genes for one or two steps cannot be identified. In these cases, prototrophy can still be predicted correctly using our 40% cutoff of missing genes. There are some bigger mysteries, like serine synthesis in *Desulfovibrio*, but these are rare and they were not considered in the current manuscript.

and provides an analysis that is based on too many assumptions, including sparse genomic data from MAGs and SAGS, to generate correlation-based results that are lacking validation.

As highlighted above, we used a conservative 'cutoff' (>40% of genes missing, see new Extended Data Figures 1 and 2) when inferring auxotrophy as we recognize that not all genes in amino acid biosynthesis pathways are known. GapMind has been shown to be accurate at correctly annotating the biosynthesis pathways of a diverse breadth of bacterial groups, and incorporates significant advances in terms of the suite of proteins and alternative pathways considered compared to widely used approaches such as CarveMe (for details, see a recent comparison of both approaches here <https://www.nature.com/articles/s41559-022-01936-3>). Note that this work (cited in the manuscript) shows that GapMind is much more accurate than CarveMe at generating annotations to infer amino acid auxotrophy in bacteria. As mentioned throughout, the MAG/SAG genomes included were heavily filtered for completeness and quality, so these are not sparse genomic data. Finally, note that the patterns identified with our study are consistent with previous work and general ecological theory, strongly suggesting the analysis is not significantly flawed by the nature of the data.

MINOR COMMENTS:

L41/L43 The authors should clearly differentiate between bioinformatic results and experimental/previous results.

We now make this clear by clarifying "Our estimates indicate..." in L41.

L67/L69 Does “outcompetition” refer to the growth rate of the auxotrophic strain in the original medium containing the amino acid?

It refers to the population size of the auxotrophic strain at the end of competition assays where both ancestral and auxotrophic strains were co-inoculated in the same medium containing all amino acids.

L76 please rephrase - Here, we focus on amino acids

Checked, thank you.

L77 please rephrase - ...amino acids can be important both

Checked, thank you.

L91 “Auxotrophy is expected to be more common ...” as the sentence is framed as speculative.

Checked, thank you.

L92 Does the 20-30% refer mainly to amino acids in proteins and/or small (poly)peptides? Also, is the term “dissolved” used to describe free amino acids? Are these single AAs or include peptides?

Excellent point. After consideration, we have decided to remove this information from the text as amino acid availability is highly variable, technically difficult to measure, and has not been thoroughly described across habitats. The amino acid proportions that had been presented here are proxies that are too broad to support hypotheses. Instead, we have decided to illustrate our expectation that auxotrophy should be more common in habitats with high availability of amino acids by exemplifying the more intuitive case of dairy products and *Lactobacillus* (L92-95).

L93 “N” was never introduced as an acronym for “nitrogen”, and the Methods section use “N” for the sample size. (Also, inconsistent use of “P” and “p” for p-value)

Checked, thank you.

L100 because of their spatial proximity?

Primarily yes, but also because these habitats might enable higher diffusion of leaky metabolites. We have amended this accordingly in L98.

L104ff the authors indicated above that their focus is on amino acid auxotrophies, but mix AA auxotrophies and other auxotrophies in the introduction text. It would help to be more explicit here, maybe use the term AA-auxotrophies to clearly differentiate the different types.

Thank you, we have made this distinction more explicit in L101.

L152 what are those 171 taxa? Is there a list that shows the phylogenetic representation of the taxa (see comment below for L159)

We now include Extended Data Table 1 as a spreadsheet that contains the taxonomic affiliation of those 171 taxa.

L159 of course – you used a manually created cutoff (40%) to achieve this accuracy..., while

this is a valid method to benchmark your approach, it is still arbitrary and could be biased (based on phylogeny, genome size and quality). The validation is performed on only 19 taxa. These 4 phyla are heavily biased towards Bacillota, aka Firmicutes (11/19 of the members belong to this phyla). The other members are 5 Proteobacteria, 2 FUSEOBACTERIOTA and 1 Verrucomicrobiota.

Thanks for pointing this out. Note that, since we took a conservative approach to avoid the overestimation of auxotrophy, we used the 40% missing genes cutoff based on the validation using 171 known prototrophs, not on the 19 known auxotrophs. To further address this point, we now include a more detailed comparison of the accuracy at predicting amino acid biosynthesis capabilities using different cutoffs (Extended Data Figures 1 and 2). These analyses show that for the correct prediction of prototrophy, a 40% cutoff clearly outperforms the 30% cutoff, while it does not do worse than a 50% cutoff. The cases of the two specific amino acids (cysteine and serine) where there are still a few incorrect predictions (only ~4% of cases), are also explicitly discussed in L159-174.

See that for the validation on the 19 taxa with known auxotrophies the overall gain in accuracy by using a 30% or lower missing genes cutoff is rather small compared to the 40% cutoff (Extended Data Figure 2). Together, these analyses indicate that, given our current knowledge of amino acid biosynthesis pathways, our approach is valid for inferring amino acid auxotrophy in bacteria.

It is also curious why *E. coli*, *B. subtilis* or other model organisms are missing – are there no experimental validation for those microbes?

Wild-type E. coli strains are prototrophic and therefore not useful for validation purposes. In the case of *B. subtilis*, the best-studied strain (strain 168) is a tryptophan auxotroph, due to a recent frameshift (that likely occurred during laboratory cultivation). Since this is a recent mutation with just 1 gene lost as a product of cultivation, the relevance of this to predicting auxotrophy in natural *B. subtilis* strains and therefore its value for validation is unclear.

L203 one study – two references?

Thank you, we have removed the incorrect citation.

L205 unfortunately – after all... it is just an “estimate”, and the authors argue that there estimate is superior over previous estimates (using an arbitrary cutoff of 40% genes missing)

See our answer to the point raised above for L159.

L222 ...group, which is...(comma)

Checked, thank you.

L235 why are bacterial families referenced here? What information from the references is being used in the discussion of the results?

Thank you for this comment. We included these references as we thought that not all readers would be familiar with these families, and the cited references include information on auxotrophy within these families so that non-expert readers can put these results into context.

L242 given the difference in auxotrophies from MAGs/SAGs and isolate genomes, I would strongly argue against the use of any MAGs/SAGs in the analysis! SAGs in particular are of low completeness (and SAGs and MAGs from the same samples seldomly match...)

Thank you for this comment. Note that we have already taken significant steps to filter out genomes with low quality and completeness. Also, we only kept those MAGs/SAGs with no chimerism and which contained an assembled 16S rRNA gene. Note that, for example, the habitat-based analysis is based almost entirely on genomes derived from isolates (>95%) and the patterns are clear (Extended Data Figure 8). More specifically, among the 4 habitats with the highest prevalence of auxotrophies, only 2.5% of the genomes were derived from MAGs/SAGs. This confirms that the patterns observed across habitats cannot be driven by potential quality issues of MAGs/SAGs. To further address this reviewer's point, we now report the weak correlations observed for the relationship between genome completeness and the number of auxotrophies per genome in lines 278-281 – see figure below.

L247 “higher minimal doubling times” for uncultivated taxa? Were these doubling times predicted? How? Approaches based on replication rates (e.g. Brown et al Nat Biotech 34, 2016) have been proven to be highly variable and not providing consistent data across different samples, containing different taxa

We obtained these minimal doubling times from the predictions of the package gRodon, which is based on quantifying codon usage bias in highly-expressed genes (see <https://doi.org/10.1073/pnas.2016810118> for details). We have now amended the sentence to clarify these were predicted minimal doubling times. We agree that estimates based on replication rates are tenuous at best and sample-specific. See L267.

L248 cultured isolates can be from different media/culture conditions. The result of a statistical test is invalid if differences in culture conditions were not controlled

As noted above, we are referring to predicted minimal doubling times based on codon usage bias (see thorough discussion and validation of this approach in the cited <https://doi.org/10.1073/pnas.2016810118>), so any considerations about media/culture conditions are not relevant to the statistics presented here.

L251 a (negative) significant relationship with GC might be anecdotal rather than a causation. Gene duplication and long intergenic regions can significantly modify the GC content of a genome

Thanks for pointing this out. We agree there is a general correlation between GC content and genome size, so we have removed this analysis from the study.

L253 The authors state that observed auxotrophies in MAGs/SAGs are likely due to genome size, a potentially misleading statement that the genome size is a cause of auxotrophy rather than a (weak) correlation. Also, the method to determine the correlation seems not appropriate for discrete data (number of auxotrophies) versus continuous data (genome size)

Thank you. We agree that sentence was potentially misleading, and we have rephrased it to clarify that it is not genome size *per se* that causes MAGs/SAGs to generally be more auxotrophic, but the underlying evolutionary processes for genome reduction (which we further examine later on in the study). See L273-274.

As discussed above, Pearson correlation is a test that does not make assumptions about the continuity of the data and is valid for this analysis.

L255 what does “complete” mean in this context? More prototrophic?

“Complete” here refers to genome completeness. We have clarified this in L275-276.

L255 completeness of MAGs is also not “exact science” but based e.g. on marker genes presence/absence (see e.g. CheckM). Completeness of MAGs has recently been linked to the recovery of functional signal (10.1038/s43705-023-00221-z), so I would argue that auxotrophy assignment is indeed linked to completeness bias

We agree that assessing genome quality and completeness is not trivial, and we have been careful to filter out all genomes that were <95% complete, had signs of chimerism, or that lacked an assembled 16S rRNA gene. Please note again that in our habitat-based analysis, the 4 habitats with the largest proportion of amino acid auxotrophs had >97% of the representative genomes derived from isolates (Extended Data Figure 8). This indicates that the patterns we are observing cannot be driven by completeness bias from MAGs/SAGs. Also, to address that issue we now report the weak relationship between genome completeness and number of auxotrophies (lines 278-281).

L258 Extended Data Fig 4 is not very meaningful (i.e. percent). Representation of absolute numbers would be better suited here

Thanks for pointing this out. We now report the absolute numbers on this graph and in Extended Data Figure 8.

L261 general statements including entire taxa are highly speculative and thus not very meaningful

Thanks. There is ample knowledge on how growth rates and nutrient requirements can impair bacterial culturability in general (see e.g. <https://www.pnas.org/doi/10.1073/pnas.2016810118>), so we think our statement is supported by the literature.

L264 "larger genomes compared": Average size of genomes might also be biased towards genomes with gene copies of unrelated pathways

Thanks, we have amended the sentence accordingly. See L288.

L270ff is this maybe because some biosynthesis pathways require less genes and thus the

40% cutoff is heavily biased? L286 shows a test for tryptophan (with a not very strong r and p value), why was this not done for all AAs?

Note that in L286-287 of the submitted manuscript we reported that the number of auxotrophs predicted for a given amino acid and the number of genes in the biosynthesis pathway of that amino acid were not significantly correlated. This means the 40% cutoff is not biased towards predicting auxotrophy for amino acids with fewer genes in their pathway. The case of tryptophan is unrelated to this point: we showcased tryptophan auxotrophy in the context of particular phyla being more auxotrophic for specific amino acids as an example that is consistent with the literature. We did not perform this analysis for all amino acids - the case of tryptophan was simply used to exemplify an association between a phylum and a specific amino acid auxotrophy that is consistent with the literature.

L284 “unsurprisingly” – please rephrase, it looks like the authors refer to gut-associated bacteria rather than to the specific genera

Thank you, we have rephrased this statement as “notably” in L307.

L286/L287 A correlation between number of genes in a certain pathway and the number of auxotrophies is unnecessary and derives from an arbitrary definition of pathway

As stated in our response above, we think this analysis is worth including as it rules out the possibility that our missing genes threshold has a strong impact on the likelihood of predicting auxotrophy depending on the number of genes involved in a given biosynthesis pathway. Also, GapMind defines a pathway based on steps starting from central metabolites as described in MetaCyc (which contains information on all experimentally described metabolic pathways), so it is not based on an arbitrary definition of pathway.

L289 “a previous work” or “previous works”?

Checked, thank you.

L289/L294 The authors should mention the coefficient value of the correlations found in previous works. Also, the correlation might be misleading, as the correlation can be associated also to the total number and type of bacteria analyzed previously

The objective of this paragraph was to show that, overall, we did not detect a significant correlation between auxotrophy and biosynthesis cost, so comparing correlation coefficients would not help to show this result more clearly. Moreover, as the reviewer points out, the strength of the correlation can vary for a number of reasons, so reporting coefficients is not very useful. The important point we want to make is that our results contradict results reported previously as we see no correlation between biosynthetic costs and amino acid auxotrophies.

L309 Fig 2 shows that auxotrophy matches abundance of Firmicutes – to rule out any bias, I would expect experimental validation of some key bacteria for the Lactobacillaceae and Streptococcaceae. Also, see L159, the algorithm was benchmarked on data that contained ~60% Firmicutes.... Coincidence?

To reiterate our responses from above - the accuracy of GapMind in specific phyla like the Firmicutes depends on the prior knowledge of the amino acid biosynthesis pathways in those phyla, not on the fact that we benchmarked our threshold predominantly on a particular phylum. The central point of our conservative approach was to set a threshold that correctly predicts known prototrophic taxa (171 strains), where only 9/171 belong to the

Firmicutes, not based on the 19 taxa with known auxotrophies (otherwise, it would not be conservative and would incorrectly infer auxotrophies more than prototrophies). Importantly, the Firmicutes is one of the best studied phyla in terms of metabolism. The Lactobacillaceae and Streptococcaceae are also very well-studied families (see e.g. <https://journals.asm.org/doi/10.1128/aem.70.3.1869-1873.2004> or <https://journals.asm.org/doi/10.1128/jb.183.24.7354-7364.2001> and citations within), so we think additional experimental validation on these groups is not justified.

L316 Extended Data Fig 6 – why only three specific proteobacteria (alpha, beta, gamma)? Why not all proteobacteria? Or is there supposed to be a separation between the different proteobacteria? The graph needs a suitable legend to explain what is shown (i.e. statistics, lines, dots).

We show auxotrophy predictions for these groups as the amino acid biosynthesis pathways for these groups are better studied than other groups within the Proteobacteria (see Figure 5 in <https://journals.asm.org/doi/10.1128/msystems.00291-20>, which shows the number of known knowledge gaps in amino acid biosynthesis across bacterial phyla). We have included the missing information on the figure caption – thanks for noticing that.

L329 How can this be shown by looking at percentages? Wouldn't this have to be checked for what organisms are preferably from MAGs/SAGs? For example, if all Firmicute data comes from MAGs/SAGs, the overall percentage shown in Extended Data Fig 8 would not be suitable to make this statement. In the next sentence the authors try to address this (reporting a poor r and p value) – indicating that in fact the analysis is biased toward the proportion of representative genomes.

Thank you. The proportion of representative genomes that belong to MAGs/SAGs in the analysis across habitats is very low (<5%) and evenly distributed across habitats, ruling out the possibility that this affected the patterns observed. See for example the case of sourdough, which contains the second highest mean number of auxotrophies per genome, but all genomes recovered in this habitat are derived from isolates (now Extended Data Fig. 8). We now report absolute numbers on the graph to address this point.

L329 Extended Data Fig 8 comes before Extended Data Fig 7 in the text

Thanks for pointing this out, which we have now corrected.

L352 It was stated in the introduction that available proteins do not necessarily translate into available/dissolved amino acids. How is this relevant in this context, but not before?

Although the proportions of available amino acids probably vary between habitats, it is reasonable to assume that protein-rich environments such as fermented foods likely contain higher amounts of available amino acids than environments that are not protein-rich. We have modified the Introduction to better explain why we expect that auxotrophy should be more common in habitats with high availability of amino acids (L92-95).

L354 what is “stable”? These statements are way to general to be of any meaning to the reader (especially since you are referring to very broad habitat classifications)

Apologies if the concept was misleading. We refer to stable as having low temporal variability, which we have clarified in the revised manuscript. See L385.

L356/L373ff instead of providing yet another genome-centric analysis, why not confirm some of these predictions?

As mentioned repeatedly in the text, validating auxotrophic predictions is highly challenging in the laboratory, and even more so under field conditions (this is why there are so few confirmed auxotrophs with available genomic data). We have provided ample evidence from the literature and the specific cases of the Bdellovibrionaceae and *Candidatus Udaeobacter* which are consistent with the auxotrophy predictions provided here, and there is also ample evidence for auxotrophy in several Firmicutes families, so it is unclear what additional experimental confirmation would add to the study. As mentioned earlier, the aim of this study is to provide general predictions on bacterial amino acid auxotrophy, not to improve an existing genome annotation tool.

L390ff ...which has been shown before as you correctly state. The entire paragraph does not provide any new insights but reiterates current knowledge. I am not sure what is gained here.

To our knowledge, no previous study has actually provided a comprehensive analysis connecting auxotrophy, genomic features, and general life history traits in bacteria. Most evidence came from disconnected studies that focused on very specific taxonomic groups (such as the case of *Pelagibacter ubique*). Overall, previous work has suggested that auxotrophy is associated with genome streamlining and we now provide comprehensive evidence to support this formerly anecdotal observation.

Methods:

L451 Is "medium-confidence" classification provided by the software, or did the authors use a threshold value to filter results from GapMind? Also, can you please provide the computer characteristics and typical run time per genome?

We followed the "medium confidence annotation" of any given metabolic pathway step defined by the original GapMind framework (<https://journals.asm.org/doi/10.1128/msystems.00291-20>). Further details on GapMind are available in the cited publication. We used a computer with 32 cores and 256 Mb RAM, which ran at 10 cpu per genome, and took 24h to annotate the >26,000 genomes. We have made available the code and data to run GapMind across GTDB genomes on figshare.

L453 The second criterion "at least 30% identity and 80% coverage and more similar to proteins with this function than to any other characterized protein in a database of >100,000 characterized proteins" lacks details such as the source of this database (GapMind uses 6,284 experimentally characterized proteins) and what is "similarity"/"more similar" as the authors use identity and coverage.

Thanks for these comments. In the GapMind framework, candidates for steps are identified by running ublast against a database of 1,849 proteins that are experimentally shown to be involved in amino acid biosynthesis. The similarity check uses a database of 113,704 experimentally-characterized proteins. "More similar" is based on the alignment's bit score to either the proteins involved in amino acid biosynthesis or the proteins that perform other functions. Based on the highest bit score, the algorithm identifies if the candidate gene is likely part of the biosynthesis pathway for any given amino acid. We have revised this section to improve clarity and precision (see L472-482).

L483/L495 Why is validation of the predictions important? As far as I understand, the authors used published software... It is not clear what is different from the previous validation, or if this new validation step introduced biases in the analysis rather than quantifying them.

GapMind is a gene annotation tool, not a predictor of amino acid auxotrophy. To make the annotations produced by GapMind useful for predicting auxotrophy, this validation is critical. Doing this validation was needed to identify the percentage of genes that had to be missing in the biosynthesis pathway of any given amino acid to accurately predict auxotrophy. Without this validation, advancing from annotation (GapMind) to prediction of auxotrophy (current study) would not be possible. Our validation based on 171 known prototrophs clearly shows that our approach infers that a taxon is auxotrophic for at least 1 amino acid correctly in ~95% of the cases – see L185.

L497/L512 This section describes the use of eggNOG-mapper to (re-)annotate the genomes and obtain COG classification, and normalize the prevalence to the genome sizes. However, it is not clear why the authors did not compare results from eggNOG-mapper and GapMind.

As mentioned above, the framework of eggNOG-mapper is not suitable for the annotation of amino acid biosynthesis pathways as its annotation database is limited to databases that classify proteins based on broad functions and protein families (such as COG or Pfam). GapMind's annotation database is tailored to specifically identify candidate genes involved in amino acid biosynthesis, and that is why it is a useful tool for the purpose of this study.

Also, it describes the use of doubling times, but it is not clear if the authors used the gRodon software, or they used data from the EGGO database. If gRodon, the authors should describe briefly the procedure to run the software. If EGGO, the authors should describe how they matched genomes from GTDB to the data available in EGGO.

Thanks for pointing this out. We have clarified that we matched the genome accessions in our study to those of the genomes with predicted minimal doubling times in the EGGO database in L545-546.

Finally, it is not clear why >3000 taxa per phyla are necessary for the statistical tests.

Thank you for this comment. We clarified that having >3000 taxa in these phyla was important to ensure that we focused our analyses on phyla that are relatively well-represented in genome databases. See L537-538.

References:

Please assure to italicize species names and not to mix different styles (e.g. all first letters capitalized)

Thank you, we have corrected the reference list.

Figures:

Figure 1 Style between Fig 1A and 1B is different. Specifically, 1B hides the stratification of the data. It is suggested that Fig 1B style be identical to Fig 1A. The scale between Fig 1A and 1B is different. Also, Fig 1A shows labels on top (a, b, etc) that are not explained. The numbers next to the taxon names in Fig 1C lack explanation as well. Why are they different from Fig 1A?

Thanks for the suggestion. We have corrected the scales. However, displaying the points on this small panel for 3613 and 22,664 unique assembled and isolate genomes compromises the visualization of these results. This was not the case in panel A. The violin plot provides more information than a simple boxplot, and clearly shows the differences in the number of auxotrophies per phylum between assembled and isolate genomes. The numbers in brackets in panel C are different because these are the subset of taxa that were auxotrophic for at least 1 amino acid. We have revised the caption to make this clear.

Figure 2 Top panel of Fig 2A is redundant with red diamonds in the bottom panel.

We decided to keep the top panel because it illustrates more clearly the range of mean auxotrophies predicted within each of the habitats, which is not obvious with the red diamonds. We keep the diamonds on the main panel because otherwise there is no way to understand the order of the x axis.

Also, letters referring to “statistical differences” don't show what pairwise samples were compared to obtain these differences.

Like in Figure 1A, these letters depict the comparisons to the neighboring groups.

In Fig 2B, some ASVs cannot be assigned to a phylum. Is the proportion of unassigned ASV low or zero?

Since, as explained in the methods section, this analysis was based on ASVs with a representative genome in any of the 12 phyla studied here, all ASVs that had less than 99.6% similarity of their 16S rRNA gene sequence to that of a representative genome were excluded from this analysis. These proportions are shown in Table 1.

Can the authors provide their abundance?

Thanks for the suggestion. Yes, we now provide these abundances in Table 1.

Also, 10 phyla were identified, while in Fig 1A and C, 12 phyla were analyzed. Was the abundance of the missing 2 phyla too low?

These samples did not contain ASVs that had a sufficiently similar match of their 16S rRNA gene to any representative genome of the phyla Spirochaetes or Tenericutes, so only 10 phyla were identified in these habitats.

Figure 3 Significant level was defined as p-value less than 0.01, that is an atypical value. No explanation of the defined p-value was provided, as it is expected that lowering the threshold for the significance, it will reduce the number of categories that are significantly different.

We chose a more stringent p-value threshold to ensure we only reported trait associations that were strongly supported with the data. Choosing a lower threshold is a widely accepted approach to avoid reporting spurious associations to broad functional annotations like COGs.

Reviewer #2 (Remarks to the Author):

The authors present a different approach and perspective on an important topic in microbial ecology. As the authors lay out in the manuscript there are many gaps in our understanding of the prevalence of bacterial auxotrophy and it is challenging to use genomic approaches to confidently identify the absence of pathways. I found the authors' conservative approach to

predicting amino acid auxotrophy and the manner in which they validated their results very thorough. While their approach identified fewer bacterial auxotrophs than other approaches, many of the trends in taxa and habitat where auxotrophs are most prevalent were consistent with previous work. I have only a couple minor comments and questions below. The paper was clear and well written.

L160: Do the authors have any ideas as to why their model did not perform quite as well with serine and cysteine? Are there more possible synthetic pathways for arriving at these particular amino acids? It seems helpful to discuss the reasons for this to help think about how to approach predicting auxotrophy for other metabolites beyond amino acids.

Thanks for the suggestion. We have done additional research and now include a discussion on why we generally overestimated the prevalence of auxotrophy for cysteine and serine in our validation. The biosynthesis pathways for both cysteine and serine contain enzymes (cysE, serC) which are poorly described and are challenging to annotate, but we actually found that all the prototrophic taxa we incorrectly inferred to be auxotrophic for cysteine and serine are bacteria with sulfur metabolisms, which are known to be able to produce cysteine and serine from pyruvate and other central metabolism intermediates (L159-174). These pathways were not accounted for by our annotation tool. However, all but one of the mentioned misclassified cysteine and serine auxotrophs belong to the phyla Desulfobacteria and Aquificae, which were not included in the main analysis because we could not retrieve more than 100 quality genomes from these phyla. This means the potential misclassification of cysteine and serine auxotrophy in our study is unlikely to influence our analyses. Besides additional discussion on this matter, we now include Extended Data Figures 1 and 2 which show how our predictions would change if we were to modify the 40% missing genes cutoff for inferring auxotrophy. These new analyses confirm this cutoff offers the optimal balance between over- and underestimating the prevalence of auxotrophy.

L500-502: I'd suggest mentioning that you required 2 auxotrophies to count a taxa as an auxotroph in the results & discussion. I think readers will assume that a single auxotrophy would be counted as an auxotroph and might miss this information in the methods section.

Thanks for the suggestion, we now include a sentence in L419-420 to include this information in the Results and Discussion.

I also think it would be worth mentioning in the methods or extended data how many more auxotrophs you would have if you did not require 2 auxotrophies. I'm curious how much this changes the predictions. I understand that the authors were trying to be conservative in the identification of auxotrophs, but this seems like it could change the results quite a bit. But, maybe it doesn't. Are most experimentally identified auxotrophs auxotrophic for more than one amino acid?

Great point. This information is available in Extended Data Table 3, where we reported the proportion of taxa within each phylum that had only 1 amino acid auxotrophy.

Reviewer #3 (Remarks to the Author):

Ramoneda et al. present a well-written analysis of the prevalence of amino acid auxotrophy across 12 bacterial phyla and study auxotrophy patterns across environments using a validated model to improve on previous estimations of amino acid auxotrophy in bacteria.

I recommend this manuscript for publication with one major and a few minor revisions.

MAJOR

The authors use an arbitrary 40% cutoff for the auxotrophy designation throughout their manuscript despite the fact that their data show this cutoff is reliable for some amino acids and too high for others such as serine and cysteine, which they acknowledge (see Extended data figures 1A and 1B, which show the number of false positives). This is a missed opportunity to use the information they have already presented to determine a tailored cutoff for each amino acid to improve accuracy. Doing so is especially important given that the authors use their data to estimate the percentage of all (sequenced) bacteria that are prototrophs.

Thank you for this comment. Indeed, we have done additional research and now include a discussion on why we generally overestimated the prevalence of auxotrophy for cysteine and serine. See our response above and the revised text (lines 159-174) where we explain this phenomenon, why we incorrectly inferred cysteine and serine auxotrophies for sulfur bacteria, and how the potential misclassification of cysteine and serine auxotrophy is unlikely to influence our conclusions (essentially because it affected phyla with sulfur metabolisms which were not included in our analyses).

There are important reasons why tailoring a specific cutoff for each amino acid will not lead to improved predictive accuracy based on the currently available data. Amino acid biosynthesis pathways are not equally well described across taxa, as the example discussed above with sulfur bacteria indicates. Based on the 19 taxa with experimentally defined auxotrophies we were able to find, one could tailor a cutoff for each amino acid, but this cutoff would be strongly biased towards those taxa (11/19 are Firmicutes). This means there is currently not enough experimental information on bacterial amino acid auxotrophy to establish tailored cutoffs for each amino acid or taxon. Still, we now provide additional quantitative evidence showing how the validation results would change if we were to increase or decrease the missing genes cutoff (see new Extended Data Figs. 1 and 2). These analyses show that for the correct prediction of prototrophy, a 40% cutoff clearly outperforms the 30% cutoff, while it performs similarly to a 50% cutoff. For the validation on the 19 taxa with known auxotrophies the overall gain in accuracy by using a 30% or lower missing genes cutoff is rather small compared to the 40% cutoff (Extended Data Figure 2). Together, these analyses indicate that, given our current knowledge of amino acid biosynthesis pathways, the conservative approach taken here is valid for inferring amino acid auxotrophy across bacterial groups.

As discussed in the manuscript, obtaining such experimental information is currently challenging because one needs to perform extensive media testing coupled with whole genome sequencing, efforts that would improve metabolic pathway annotation tools but are outside the scope of this study. Note that GapMind is based on important improvements regarding the description of amino acid biosynthesis pathways based on similar efforts (<https://journals.plos.org/plosgenetics/article?id=10.1371/journal.pgen.1007147>; <https://journals.asm.org/doi/10.1128/msystems.00291-20>). There is also a recent publication showing the improvement in genome annotations provided by GapMind over the widely used CarveMe (<https://www.nature.com/articles/s41559-022-01936-3>).

MINOR

General:

The words “Ecological perspective” in the title were misleading, I thought the manuscript was going to be a hypothesis piece rather than a research article. Should be retitled to capture everything that they did more specifically.

Thanks for the suggestion. We agree that the word “perspective” could be confusing to readers and we have now revised the title.

The authors allude to the cultivability bias in their dataset, but it should be acknowledged explicitly.

Thank you. We agree that there is a better knowledge of the metabolism of cultured taxa that could have affected our predictions. This is now mentioned on lines 188-191.

Figure 1A and 2A captions: Explain what the diamonds are

We have added this information - thank you for noticing that.

The authors should explain why only 17 amino acids were chosen and why chorismate was included in the analysis.

Following GapMind’s framework, alanine, aspartate, and glutamate were excluded from the analysis because these can be produced via transaminase reactions from intermediate compounds from central metabolism, and transaminase substrates are very challenging to annotate which would lead to many low-confidence hits. We now state this in L487-492.

Extended data figures 7 and 8 are not mentioned in order in the text.

Thanks for noticing this, we have corrected the order.

Specific:

91-96 should include numbers for total N in addition to amino acid percentages since availability will depend on total amino acids. Lines 353-356 contradict the statement about marine and freshwater environments.

Indeed, as noted above we have decided to remove this information as amino acid availability is highly variable, technically difficult to quantify, and has not been thoroughly described across habitats. Instead, we have decided to illustrate our expectation that auxotrophy should be more common in habitats with high availability of amino acids by exemplifying the more intuitive case of dairy products and *Lactobacillus* (L92-95).

103 shorten sentence to improve readability

Checked.

127 contains a typo “are remain”

Checked.

199 The definition of taxa here is unclear, is this referring to individual genomes?

We now specify that each taxon is represented by a single genome.

219 It is unclear what the expectation was for Cyanobacteria

We make explicit the expectation that all Cyanobacteria are able to synthesize all amino acids and include a more appropriate reference to support this expectation. See L238-239.

245 what proportion of MAGs / SAGs were from uncultivated taxa?

Based on NCBI's information on isolation source, 95.8% of the MAGs/SAGs included in our analysis do not have an isolation source. See L266.

250 The authors explain the relationship between auxotrophy and genome size but do not explain the GC content relationship.

Thanks for pointing this out. The relationship with GC content is most likely due to its co-variation with genome size, so we have decided to remove this from the study to avoid confusion.

254 It is unclear what the percentages in parentheses are for

We now specify that these percentages indicate average genome completeness.

345 "Select for" is too strong, please rephrase it to something like "Support"

We agree and have rephrased accordingly.

377 typo: should refer to fig 3C, not 4C

Checked.

435 How was genome completeness estimated?

We now specify that genome completeness was estimated using CheckM and include the corresponding reference in L467.

Sincerely,

Josep Ramoneda and Noah Fierer (on behalf of all co-authors)

Cooperative Institute for Research in Environmental Sciences (CIRES)

Department of Ecology and Evolutionary Biology, University of Colorado Boulder (USA)

Reviewer #2 (Remarks to the Author):

The authors addressed all my comments and questions from my first review.

Reviewer #3 (Remarks to the Author):

The authors have adequately responded to our queries and we have no further concerns, except that the revised title still does not reflect the content of the manuscript, as the manuscript does not directly address ecology. The authors could use a term such as "computational prediction" and perhaps refer to the new finding that auxotrophies are less widespread than previously thought.

October 23, 2023

Please see below our responses to the comments raised by two independent reviewers of this work.

Reviewer #2 (Remarks to the Author):

The authors addressed all my comments and questions from my first review.

We would like to thank once again the reviewer for the constructive feedback on this manuscript.

Reviewer #3 (Remarks to the Author):

The authors have adequately responded to our queries and we have no further concerns, except that the revised title still does not reflect the content of the manuscript, as the manuscript does not directly address ecology. The authors could use a term such as "computational prediction" and perhaps refer to the new finding that auxotrophies are less widespread than previously thought.

Indeed, we agree with the reviewer that the new title still does not cover the content of the work. Following the advice from both the reviewer and the editor, we now provide the new title "The taxonomic and environmental distributions of bacterial amino acid auxotrophy", which more accurately reflects the content of the work.

Please do not hesitate to contact us for additional information or any arising matters regarding this manuscript.

Best regards,

Josep Ramoneda & Noah Fierer (on behalf of all co-authors)

Cooperative Institute for Research in Environmental Sciences (CIRES), University of Colorado
Boulder (USA)

Department of Ecology and Evolutionary Biology, University of Colorado Boulder (USA)